# Seasonal Significance of New Particle Formation Impacts on Cloud Condensation Nuclei at a Mountaintop Location

Noah S. Hirshorn[1], Lauren M. Zuromski[1], Christopher Rapp[2], Ian McCubbin[1], Gerardo Carrillo-Cardenas[1], Fangqun Yu[3], A. Gannet Hallar[1]

[1] Department of Atmospheric Sciences, University of Utah, Salt Lake City, 84112, United States
[2] Department of Earth, Atmospheric, and Planetary Sciences, Purdue University, West Lafayette, 47907, United States
[3] Atmospheric Sciences Research Center, State University of New York, Albany, 12203, United States

*Correspondence to*: A. Gannet Hallar (gannet.hallar@utah.edu)

**Abstract.** New particle formation (NPF) events are defined as a sudden burst of aerosols followed by growth and can impact climate by growing to larger sizes and under proper conditions, potentially forming cloud condensation nuclei (CCN). Field measurements relating NPF and CCN are crucial in expanding regional understanding of how aerosols impact climate. To quantify the possible impact of NPF on CCN formation, it is important to not only maintain consistency when classifying NPF events but to also consider the proper timeframe for particle growth to CCN relevant sizes. Here, we analyze 15 years of direct measurements of both aerosol size distributions and CCN concentrations and combine them with novel methods to quantify the impact of NPF on CCN formation at Storm Peak Laboratory (SPL), a remote, mountaintop observatory in Colorado. Using the new automatic method to classify NPF, we find that NPF occurs on 50% of all days considered in the study from 2006 to 2021 demonstrating consistency with previous work at SPL. NPF significantly enhances CCN during the winter by a factor of 1.36 and the spring by a factor of 1.54, which, when combined with previous work at SPL, suggests the enhancement of CCN by NPF occurs on a regional scale. We confirm that events with persistent growth are common in the spring and winter, while burst events are more common in the summer and fall. A visual validation of the automatic method was performed in the study. For the first time, results clearly demonstrate the significant impact of NPF on CCN in montane North American regions and the potential for widespread impact of NPF on CCN.

## 1. Introduction

Atmospheric aerosols, which originate from primary emissions or through secondary gas-to-particle conversions, are a large source of climatic uncertainty (Stocker et al., 2014). Aerosols can affect Earth's radiative balance directly by interacting with incoming radiation, and indirectly through their role as cloud condensation nuclei (CCN) (Twomey, 1974; Twomey et al., 1984; Albrecht, 1989; Charlson et al., 1992). New particle formation (NPF) is a source of atmospheric aerosols that involves the formation of particles less than 3 nm in diameter and the subsequent growth of these freshly nucleated particles to larger sizes (Yu and Luo, 2009; Spracklen et al., 2010). These secondary aerosols originating from NPF can indirectly impact climate by acting as CCN (Kerminen et al., 2012; Gordon et al., 2017; Kerminen et al., 2018). Previous modeling studies

estimate that the contribution of secondary aerosols from NPF to CCN is significant in the free troposphere, with some estimates predicting that 35% of CCN, at a supersaturation of 0.2%, originate from secondary aerosols (Merikanto et al., 2009). Gordon et al., (2017) estimates that at a supersaturation of 0.2%, 67% of CCN at low-level cloud heights in the pre-industrial atmosphere are attributed to NPF, compared to 54% in the current atmosphere (Gordon et al., 2017).

Understanding the contribution of NPF to CCN in clean, remote environments will not only expand the regional understanding of how NPF can impact CCN but also will allow for potential estimates of CCN concentrations in the pre-industrial atmosphere, providing a baseline for the comparison of current, anthropogenic-influenced climate trends to those of the pre-industrial atmosphere (Carslaw et al., 2017).

Mountaintop studies evaluating the relationship between NPF and CCN are important in understanding the impact that NPF can have on CCN in remote regions, including the free troposphere (Hallar et al., 2011; Hallar et al., 2016; Rose et al., 2017; Sellegri et al., 2019). A review by Zhu et al., (2021) analyzes data from multiple campaigns at Mt. Tai in China showing that NPF does contribute to CCN at the site, but decreases in anthropogenic sulfur dioxide ($SO_2$) over time have contributed to lower production of CCN number concentrations from NPF events (Zhu et al., 2021). At the Mt. Chacaltaya Observatory,

Rose et al., (2017) finds that 61% of NPF events during a 2012 study grew to CCN relevant sizes reaching a minimum diameter of 50 nm – 150 nm, and that these events are highly likely to enhance CCN concentrations, especially in the free troposphere (Rose et al., 2017). Because a CCN counter was not available at Mt. Chacaltaya, the study utilizes a methodology to identify times in which NPF contributes to CCN concentrations, starting from when aerosol number concentrations at 50 nm, 80 nm, and 100 nm begin to increase and ending when the maximum number concentration is

observed at the respective size bin (Rose et al., 2017). Recent observations from a remote site in the western Himalayas estimated the survival probability to show that a majority of secondary aerosols grew to CCN relevant sizes during observations; there was an 82% probability that a particle would grow to 50 nm and a 53% probability that a particle would grow to 100 nm (Sebastian et al., 2021; Pierce and Adams, 2007). Findings from these mountaintop studies show the potential of secondary aerosols to activate as CCN at clean mountaintop sites, but also highlight the importance of long-term

studies at different remote locations to increase regional understanding of how NPF impacts CCN in different environments since the ability for secondary aerosols to impact CCN is highly dependent on the regional characteristics of a given observatory.

Observational studies that consider the contribution of NPF to CCN formation must classify NPF events and determine the

time at which CCN is enhanced by NPF. Historically, NPF is classified visually by scientists looking for a particle burst and subsequent growth over multiple hours, forming a new nucleation mode (Dal Maso et al., 2005). However, visual classification can lead to potential human biases and brings into question the accuracy of comparisons between studies (Joutsensaari et al., 2018). An automatic methodology to identify NPF, such as the one used in this study, can minimize human bias during long-term studies at different locations. Since the time period in which CCN is enhanced by NPF is

highly dependent on particle growth, the time to reach CCN relevant sizes can range from a few hours in polluted environments with high growth rates, to over a day in remote environments with lower growth rates (Kerminen et al., 2012).

In an effort to increase the number of observational studies relating NPF to CCN, previous studies, both with and without a CCNC, have developed various methodologies to determine the time period in which observed CCN concentrations can be
attributed to the occurrence of an NPF event (Kalivitis et al., 2015; Kalkavouras et al., 2017; Dameto de España et al., 2017; Rose et al., 2017; Kalkavouras et al., 2019; Kecorius et al., 2019; Rejano et al., 2021; Ren et al., 2021) Similar to the methodology of Rose et al., (2017), detailed above, Kalkavouras et al., (2017) estimates CCN by finding particle concentrations above the minimum size required for aerosols to activate as CCN and then considers the environmental supersaturation when estimating how many aerosols in a given distribution could act as CCN. This approach further
calculates the droplet number and considers how supersaturation, chemical composition, and updraft velocity may impact the cloud droplet number (Kalkavouras et al., 2017). An evolution of this approach by Kalkavouras et al., (2019) calculates the relative dispersion of CCN at different supersaturations and considers CCN times when the relative dispersion is higher than initial conditions before a CCN event (Kalkavouras et al., 2019). This method was further employed at 35 different sites around the globe, both urban and remote, to determine the impact NPF has on CCN concentrations (Ren et al., 2021).
Kecorius et al., 2019 utilized a CCNC in the arctic to analyze CCN enhancements by fitting a slope to CCN measurements starting when aerosol formation rates increased and ending when an air mass shift occurred (Kecorius et al., 2019). In another study utilizing a CCNC in Vienna, Austria, Dameto de España et al., (2017) considers CCN number concentrations for a time period that occurs after, and for the same duration as the time difference between NPF initiation and when particles reach CCN relevant sizes (Dameto de España et al., 2017). When it comes to determining the time period that NPF
may impact CCN for long term datasets, the methodology should not only efficiently and independently (without using CCN observations) ensure that aerosols are growing to CCN sizes but also needs to consider the growth patterns of individual NPF events to accurately determine when NPF stops contributing to CCN.

In this study, we present 15 years of aerosol and CCN data from Storm Peak Laboratory (SPL), a remote, mountaintop
observatory in Steamboat Springs, CO USA, and quantify the impact of NPF events on CCN concentrations. Datasets of this length are rare and provide a unique opportunity to quantify long-term trends that have enough data to make statistically significant conclusions. NPF occurs frequently at SPL allowing for the seasonal comparison of the relationship between NPF and CCN (Hallar et al., 2011). To identify the occurrence of NPF and when to consider CCN concentrations, we present two new, statistical-based methods: one to classify NPF events, and another to determine the period in which CCN number
concentrations can be attributed to NPF.

## 2. Methodology

### 2.1 Storm Peak Laboratory

Storm Peak Laboratory (SPL) is a remote, mountaintop observatory (3210 m.a.s.l, 40.455°N, 106.745°W) located in Steamboat Springs, CO. SPL is one of the only sites in North America with long-term measurements of aerosol size distributions and CCN number concentrations (Lowenthal et al., 2002; Borys and Wetzel, 1997; Hallar et al., 2017). The laboratory is commonly in-cloud during storms and sees frequent NPF events (Hallar et al., 2011; Borys and Wetzel, 1997; Lowenthal et al., 2019). The primary wind direction at the laboratory is westerly, which allows for the potential transport of $SO_2$ and formation of sulfuric acid ($H_2SO_4$), an NPF precursor, from multiple powerplants $50 - 250$ km upwind of SPL (Hallar et al., 2016; Obrist et al., 2008). SPL is located above a mixed forest allowing for the emission of a variety of different biogenic volatile organic compounds (BVOCs) that can impact aerosol formation and growth (Amin et al., 2012). Given the remote, mountaintop location of SPL, clean atmospheric conditions are common at the laboratory (Obrist et al., 2008).

To measure aerosols at SPL, we use a TSI Inc. (Shoreview, MN) Scanning Mobility Particle Sizer (SMPS) 3936 (with a TSI 3010 Condensation Particle Counter [CPC]) for particles with diameters between 8 and 340 nm that scans every 5 minutes. Data is collected on a log normal scale with particle diameter on a log scale and time on a normal scale. The instrument is periodically shipped back to TSI Inc. for routine maintenance and calibrations. The sheath and sample flow rates were 10 L $min^{-1}$ and 1 L $min^{-1}$, respectively, for the SMPS. Multiple charge corrections and diffusion corrections are applied to all SMPS data used in the analysis. SMPS data from SPL are now available on the EBAS database (database of European Monitoring and Evaluation Programme) including level 1 data, which maintains 5 min time resolution while removing invalid values and calibrations, as well as level 2 data which presents hourly averages and quantifies atmospheric variability. Level 1 SMPS data is used in this study. The goal of EBAS data is to store long-term atmospheric science datasets and provide standards for quality assurance, thus rigorous standards for data quality are implemented to any data admitted to EBAS (Norwegian Institute for Air Research). SPL consistently runs a single-column Droplet Measurement Technology (DMT; Boulder, CO) cloud condensation nuclei counter (CCNC) that collects number concentrations of CCN every second (Lance et al., 2006; Roberts and Nenes, 2005). We consider instrument measurements at supersaturation levels between 0.2% and 0.4% in our study.

### 2.2 An Automatic Method to Classify New Particle Formation

A crux of studying atmospheric NPF is the identification of NPF events. The identification process historically utilized three-dimensional plots of log-normal size distributions and visual inspection aimed at identifying a burst of particles below 20 nm, followed by growth over the course of multiple hours that forms a new nucleation mode (Dal Maso et al., 2005; Kulmala et al., 2012). By visually inspecting these plots, the viewer sorts days into the following broad categories based on the

observed growth patterns: event, non-event, or undefined. In an effort to improve the visual classification process proposed by Dal Maso et al., 2005, studies split events into subcategories to provide more specific classifications detailing whether

particle growth is sustained during a given day, or if the given day exhibits a burst of particles (Hirsikko et al., 2007; Kulmala et al., 2012; Boy et al., 2008; Svenningsson et al., 2008; Dal Maso et al., 2005). The visual classification of NPF can present problems since human biases can influence classification leading to issues with the reproducibility and comparability of studies (Joutsensaari et al., 2018). To minimize the potential biases that influence visual classification, we present a statistically based, automatic sorting technique that evaluates particle burst and growth patterns to classify days into

one of the following categories: type 1a event, type 1b event, class II event, undefined, or non-event (Hirsikko et al., 2007; Tröstl et al., 2016). The logic of the automatic classification technique is shown in flowchart form (Figure 1) and described below.

The first step of the automatic classification method is to ensure the availability of SMPS level 1 data. Although NPF events

can span multiple days, we consider daily data (0:00 – 23:59 MST) as well as the first 12 hours (0:00 – 12:00 MST) of the next day to ensure the consideration of an NPF event doesn't prematurely end if growth continues overnight. 5-minute SMPS data is only considered if the first 24-hour period meets the following conditions: there are at least 16 hours of data present, and the period between 10:00 – 23:00 MST (the times in which NPF is most common at SPL) has less than 1 hour of data missing.


The days that successfully undergo quality control are then considered by the automatic classification method. For data to be classified as an event, two general conditions must be met: a burst of particles in the nucleation mode, and growth that spans multiple hours contributing to the formation of a new nucleation mode. To first address the presence of a burst and identify days that are non-events, we compute the percentiles of all particle concentrations in our dataset below 25 nm from 10–23

MST. Days below the 10th percentile were automatically categorized as non-event since they are automatically assumed to not have high enough nucleation mode number concentrations for an NPF event to have occurred. For days where the average particle concentration below 25 nm is above the 10th percentile of all data considered, the maximum of the Gaussians is calculated at each size bin. The normal distributions were fit by solving for the non-linear least-squares estimates using the R programming language (Equation 1) which considers the particle size distribution at each diameter to

return the time that corresponds to the maximum concentration at that given diameter (Bates and Watts, 1988). In the equation, "k" is the maximum aerosol number concentration, "t" is the time index where the normalized maximum at $D_p$ occurs, "μ" is the mean aerosol concentration, and "σ" is the corresponding standard deviation. This equation is used for the calculation of individual maximum Gaussians at each size bin:

$$f(t \mid k, \mu, \sigma) = ke^{-\frac{(t-\mu)^2}{2\sigma^2}}, \quad k = max\left(\frac{dN}{dlogD_p}\right) \quad\quad\quad (1)$$

The derived time index represents the time at which the maximum of the peak fitted particle size distributions occurs for each value of $D_p$. For data where at least 5 different Gaussian maximum points are calculated, a linear regression is fit to these maxima allowing for further analysis of growth over the course of an event (Lehtinen and Kulmala, 2003). $R^2$ values for the linear regression (one below 20 nm and another from 20 nm to about 70 nm), as well as the time differences between the maxima, are also considered to ensure growth. For days to be defined as an event, the time difference between bin maxima must be positive and non-zero for at least 40% of occurrences, the largest $r^2$ value must be at least 0.6, there must be at least five maxima considered in the fit, and the largest size bin with a calculated Gaussian maximum must be above 25 nm for type 1a event classification and 15 nm for type 1b event classification. While 15 nm may seem like a low threshold for NPF growth, the growth of a given event often reaches sizes exceeding the diameter where the last Gaussian is calculated. Figure 2 is an example of a day that is calculated as an event because the threshold is lowered to 15 nm. Days that do not meet the above statistic-based criteria are initially classified as undefined but can be classified as a class II event later in the method.

For days that are defined as an event, the growth rate and event start and stop times are calculated. To find the growth rate, a linear regression is fit to the maximum Gaussians which are time dependent. The growth rate is determined by the following equation:

$$GR = \frac{d}{dt}\left(D_p\right) = \frac{\Delta D_p}{\Delta dt} \tag{2}$$

Because the slope of the linear regression fit of the maximum Gaussians represents particle growth over time during NPF events, this value is used when determining the growth rate. This method is most similar to the log-normal function fitting method of calculating growth rate but finds the growth rate by fitting a linear regression the maximum Gaussians. Derivatives of the linear regressions are used to determine the start and end time of events, where the start time of the event is defined by the time of the first maximum of the first-order derivative, and the end time of the event is defined by the time of the last first-order derivative minimum. Figure 3 illustrates an example of an NPF event, and a day classified as a non-event.

Days that are not classified as a type 1a or type 1b event are further considered to determine whether the given day is a class II event or undefined. Class II events are different than type 1a and type 1b events due to the presence of a particle burst which resembles an "apple" shape rather than persistent growth (Dal Maso et al., 2005; Junninen et al., 2008). Because the methodology detailed thus far considers growth patterns, significant class II particle bursts are initially classified as undefined due to weak growth (undefined stats fail) or Gaussian stacking in which greater than 75% of the calculated

Gaussian maxima occur at the same time (undefined burst). To address class II events, we apply an additional set of
threshold tests to determine if days initially classified as undefined should be classified as a class II event.

Days that were classified as "undefined burst" or "undefined stats fail" are eligible for re-classification as class II events based on multiple thresholds. Class II events exhibit Gaussian maxima that occur at elevated number concentrations, exhibit particles bursting to a larger size, and originate in the nucleation mode; however, there is often a minimal difference in the
temporal location of the calculated Gaussian maxima. Thus, the focus of this analysis is the identification of a significant particle burst. To confirm that the burst originates with smaller particles and exceeds the sizes required for class II events, the lowest size bin of a calculated Gaussian for a given day must be below 15 nm and the highest size bin of a calculated Gaussian must be above 15 nm. To ensure that the given day exhibits a strong enough burst for consideration as a class II event, at least 50% of calculated Gaussians must have dn/dlogDp values above the 95th percentile of all values in a given
day. In addition, the diameters of consecutively calculated Gaussian maxima for days initially classified as an "undefined burst" cannot differ by more than 20 nm. The reason this threshold is not applied to days initially defined as "undefined stats fail" is because there is some growth observed, thus removing days with large Gaussian maxima differences could lead to the accidental removal of a class II NPF event that exhibits weak growth in addition to a significant burst.

**2.3 Formation Rates ($J_8$) and Condensation Sink (CS) Values**

The aerosol formation rate ($J_8$) and condensation sink (CS) values are calculated as part of the automatic classification method. $J_8$ values are calculated for type 1a and type 1b events. CS values are calculated for the comparison of values between type 1a and type 1b events and non-events.

The $J_8$ value for an event is defined by the formation rate equation (Kulmala et al., 2004; Kulmala et al., 2012):

$$J_8 = \frac{\Delta N_{8,D_{max}}}{\Delta t} + CoagS_{d_p} * N_{d_p} + \frac{GR}{\Delta d_p} * N_{d_p} \tag{3}$$

Where $\Delta N_{8,Dmax}$ is the change in the number concentration of particles across the considered size distribution from about 8 nm to 25 nm during $\Delta t$ which is the time difference from the defined start of an event to the defined end of an event. When
calculating the initial and final number concentrations, we utilize the average number concentration observed between 4 hours and 1 hour prior to NPF initiation as the initial number concentration. The final number concentration is the average number concentration from all 5-min scans taken during an event. Doing so allows for the comparison of the initial conditions of an NPF event, and aerosol formation across the entirety of a given event. The additional loss terms in the equation represent loss to the coagulation sink, and loss due to growth out of the size range (Kulmala et al., 2012). The entire

size distribution measured by the SMPS is used when calculating the coagulation sink loss term (Casquero-Vera et al., 2020).

CS values are calculated for the entire size distribution using the following equation (Pirjola et al., 1999; Kulmala et al., 2001):


$$CS = 2\pi D \int_0^\infty d_p \beta_m(d_p) n(d_p) dd_p = 4\pi D \sum_i \beta_i r_i N_i \tag{4}$$

In the equation, $r_i$ is the radius of a given size bin (cm), and $N_i$ is the number concentration (#/cm$^3$) of the given size bin. $D$ is the diffusion coefficient of vapor, which is assumed to be 0.13 cm$^2$s$^{-1}$ for H$_2$SO$_4$ at SPL based on calculations using

representative pressure and temperature at the site (Hanson and Eisele, 2000; Welty et al., 2008; Tuovinen et al., 2021). $\beta_m$ is calculated following the protocols of Kulmala et al., 2012 and Tuovinen et al., 2020 where the mass accommodation coefficient in these calculations is assumed to be unity (Kulmala et al., 2001; Nishita et al., 2008; Hallar et al., 2011; Kulmala et al., 2012; Tuovinen et al., 2020).

**2.4 Determining When to Consider Cloud Condensation Nuclei Concentrations**

Determining whether an NPF event is impacting CCN concentrations is crucial in understanding the exact contribution of aerosols to cloud formation and, thus, understanding the potential climatic impacts. While environmental supersaturation and particle hygroscopicity are both crucial factors for CCN activation, aerosols from NPF must grow to CCN relevant sizes

before activating as CCN. Therefore, it is important to consider CCN enhancements due to NPF at times when particles reach CCN sizes. In this study, we propose and apply a statistical method to determine the time in which to consider the contribution of NPF to CCN concentrations. Our method sets a start and end time for CCN concentrations based only on aerosol concentration measurements that consider growth patterns of aerosols over and around the time of NPF.

For days classified as type 1a events and type 1b events, the start time of CCN consideration (CCN$_{start}$) is the first time after the start of an NPF event that 25% of all particles in a given scan (ranging from 8 nm to about 340 nm) are above 40 nm. Utilizing a percentile-based threshold method to determine CCN$_{start}$ allows for newly formed particles to grow to CCN sizes and is an effective metric when dealing with multi-year datasets. CCN$_{start}$ for non-event days is calculated by using the average of CCN$_{start}$ calculated for each season during events. We consider CCN concentrations during non-events to

determine if NPF events result in an enhancement of CCN. Sunlight is generally necessary for NPF and growth; therefore, it is important to consider the variations in the seasonal diurnal cycle and obtain one unique value of CCN$_{start}$ for each season that accurately represents the time that NPF impacts the site ~~for~~ during each season (Hallar et al., 2011).

The end time of CCN consideration ($CCN_{end}$) is determined by finding the time at which particle growth from an event tapers off. To do so, we estimate the bin that corresponds to the normalized maximum aerosol concentration at each timestamp from the start of the NPF event to 17:00 MST the next day. This ensures that consecutive events are not erroneously considered. The maximum bin diameter at each timestamp is determined in a similar way to the NPF classification method (equation 1), but when considering CCN, we find the maximum of fitted Gaussians at each timestamp. Because the formation of CCN from nucleated particles can exceed the time period of NPF, especially in remote environments, our method allows for the evolution of particle growth over a time period long enough to ensure particles originating from NPF can grow to CCN sizes.

Once each time has a corresponding diameter maximum, we evaluate the overall growth pattern by fitting a polynomial curve to the Gaussian bin maximums over time. Once the curve is fit, the time at which the last inflection point occurs (in which the fitted line transitions from positive slopes to negative slopes) is selected as $CCN_{end}$. The last inflection point of the curve serves as an indicator of growth tapering off and therefore, we assume that the enhancement of CCN from NPF has concluded. For non-events, $CCN_{end}$ is determined by adding the average duration of CCN consideration ($CCN_{end} - CCN_{start}$) to the previously averaged $CCN_{start}$. Four different values of $CCN_{end}$, one for each season, are determined when finding $CCN_{end}$ values for non-events. An example NPF day including labels of $CCN_{start}$ and $CCN_{end}$ illustrating the time at which we assume CCN is enhanced by NPF is included in Figure 3. To compare the impact NPF events have on CCN, CCN number concentrations directly measured are considered during the time period spanning from $CCN_{start}$ to $CCN_{end}$ during valid events and non-events. An average CCN number concentration for supersaturation levels between 0.2% and 0.4% is calculated for each individual time period. These values are then averaged each season separately between events and non-events. The goal is to determine whether CCN concentrations are enhanced by NPF events. During long-term studies, especially at clean, remote locations like SPL, directly comparing events and non-events will result in the relative enhancement of CCN due to events at a given location. By removing the subjectivity of selecting idealized cases, we provide a more robust methodology to evaluate long-term datasets. The methodology within this paper carefully considers similar timeframes within the diel pattern with and without NPF, to look at the relative change induced by NPF. At other high-altitude mountaintop sites around the globe, this approach could have sources of error since NPF can be associated with the transport of both condensable vapors and pre-existing aerosol that could become CCN (Sellegri et al., 2019). However, SPL seems to be an exception to this rule since previous observations of NPF show association with lower existing particle surface areas which allows for a more direct comparison of events and non-events (Hallar et al., 2011; Sellegri et al., 2019). By further comparing events to non-events through a seasonal lens, we ensure that days with similar meteorological conditions are compared. By further comparing events to non-events through a seasonal lens, we ensure that days with similar meteorological conditions are compared.

## 3. Results

### 3.1 15 Years of New Particle Formation at Storm Peak Laboratory

Over the course of 15 years (2006 – 2021), we consider 835 days that pass basic quality control protocols and have both aerosol and CCN data available for analysis. The automatic method to determine NPF classification splits the data into one of the following five categories: type 1a event, type 1b event, class II event, undefined, or non-event (Hirsikko et al., 2007; Tröstl et al., 2016). Of the 835 days considered, 95 days are classified as a type 1a event, 80 days are classified as a type 1b event, 244 days are classified as a class II (burst) event, 269 days are classified as undefined, and 147 days are classified as a non-event. When considering the overall NPF event frequency, which includes type 1a events, type 1b events, or class II events, the overall event frequency calculated by the automatic method is 50% which compares well to the 52% overall event frequency observed at SPL by Hallar et al., (2011) (Hallar et al., 2011).

Evaluating NPF from a seasonal lens at SPL creates a better understanding of how important variables, such as temperature, $SO_2$ concentrations, and the presence of organics, affect NPF (Hallar et al., 2016; Hallar et al., 2013). Table 1 details the frequency of NPF events across all seasons. The summer and fall display the highest frequency of events with either a type 1a event, type 1b event, or class II event occurring on 56% of days in the summer and 59% of days in the fall. Spring (53%) and winter (41%) display similar but slightly lower event frequencies than the summer and fall at SPL. An analysis focusing on the frequency of different event types is conducted to determine which seasons may be conducive to the occurrence of type 1a and type 1b events in which persistent growth occurs. We find that type 1a events and type 1b events are more likely to occur in the winter (62% of all NPF events) and spring (51% of all NPF events) than in the summer (17% of all NPF events) and fall (32% of all NPF events) where burst events are more common partially due to higher temperatures (Yu et al., 2015).

When analyzing the impact of NPF on CCN concentrations, it is important to focus on days that exhibit a prolonged period of consistent particle growth allowing for aerosols from NPF to reach CCN relevant sizes. While type 1a events, type 1b events, and class II events are all considered NPF events, class II events do not exhibit strong, consistent growth making it difficult to calculate growth statistics (Dal Maso et al., 2005). From this point forward, we focus on comparing type 1a events and type 1b events against non-events to better understand how aerosols from NPF affect CCN. Figure 4 compares the average number of particles of a given size produced during type 1a and type 1b events against non-events. We find that NPF days at SPL produce significantly more particles than non-event days up to diameters of 82.0 nm, which is larger than the critical diameter, theorized to be as low as 30 nm at SPL, required for aerosols to activate as CCN (Lowenthal et al., 2002). The significant increase in particles between 30.0 nm and 82.0 nm during type 1a and type 1b NPF events, providing an average enhancement of 2.78 (# $cm^{-3}$) times more particles during events, demonstrates an important influx of particles from NPF that reach sizes relevant to CCN formation at SPL. Above 82 nm, days with NPF events do not indicate more

particles than non-events, which suggests any enhancements in CCN due to NPF events are likely due to particles below 82
325  nm. Previous work at SPL has shown that during NPF events, particles as low as 5 nm are observed alongside events demonstrating that particles observed during NPF originate from nucleation (Hallar et al., 2016).

### 3.2 Enhancements of Cloud Condensation Nuclei due to New Particle Formation

To better understand the extent that aerosols from NPF events affect CCN concentrations, additional quality control is conducted to determine days when NPF events grow to CCN relevant sizes and days with available CCN data taken at supersaturation levels between 0.2% and 0.4%. If there are errors in the CCN data, or the difference between $CCN_{start}$ and $CCN_{end}$ is less than an hour, the day is discarded from CCN consideration. We compare 139 type 1a and 1b events that exhibit growth to CCN sizes against 111 non-events.

Figure 5 illustrates comparative CCN number concentrations following type 1a/1b events and non-events. We find that NPF enhances CCN concentrations by a factor of 1.54 in the spring and 1.36 in the winter. Higher CCN concentrations during NPF events than non-events are statistically significant in both the winter (p = 0.020) and spring (p = 0.025). However, CCN concentrations between events and non-events during the summer (p = 0.889) and fall (p = 0.432) are not statistically
significant. Average number concentrations of CCN are higher during NPF events in the spring (event: 146.47 # $cm^{-3}$, non-event: 94.92 # $cm^{-3}$), winter (event: 98.60 # $cm^{-3}$, non-event: 72.67 # $cm^{-3}$), and fall (event: 258.84 # $cm^{-3}$, non-event: 245.61 # $cm^{-3}$) but lower during NPF events in the summer (event: 306.63 # $cm^{-3}$, non-event: 388.05 # $cm^{-3}$).

### 4. Discussion

NPF significantly enhances CCN concentrations in the spring and winter, the two seasons with the highest frequency of type 1a and type 1b events. Previous work at SPL indicates that the increased prevalence of anthropogenic $H_2SO_4$ precursors and cooler temperatures are two potential reasons that can lead to conditions that are conducive to NPF during the spring and winter seasons (Hallar et al., 2016; Yu and Hallar, 2014). While previous laboratory studies suggest that multiple gases
including ammonia, amines, and organic compounds all influence NPF, $H_2SO_4$ is important for initiating particle nucleation due to its low volatility under atmospheric relevant conditions (Yu et al., 2015; Sipila et al., 2010). $SO_2$, which is a precursor of $H_2SO_4$, is emitted from coal-fired powerplants upwind of SPL allowing for the transport of $SO_2$ which has been previously observed at SPL and can help explain the high frequency of NPF events (Hallar et al., 2016). In addition to $H_2SO_4$, lower temperatures are another important factor that can aid the enhancement of particle nucleation by lowering the
thermodynamic energy barrier required for nucleation to occur (Yu, 2010; Bianchi et al., 2016; Duplissy et al., 2016; Lee et al., 2019). The combination of prevalent $H_2SO_4$ precursors and lower temperatures are two possible factors that can allow for the occurrence of persistent NPF on a regional scale during the spring and winter (Yu and Hallar, 2014). These results from

modeling work suggest the significant enhancement of CCN due to NPF events during the winter and spring at SPL may be applicable on a regional scale in remote regions of North America downwind of power plants providing insight into the processes that drive CCN formation.

NPF does not significantly enhance CCN concentrations in the summer and fall seasons, compared to the spring and winter seasons (Figure 5). One factor that could help explain this phenomenon are higher temperatures observed in the summer and fall compared to the spring and the winter. Higher temperatures in the summer and the fall, the seasons where NPF is not significant for forming CCN, can be a barrier to nucleation since higher temperatures lead to lower supersaturation ratios of $H_2SO_4$ (Yu et al., 2015). In addition to higher temperatures, previous work shows that $SO_2$ concentrations at SPL are slightly lower in the summer and fall than in the spring and winter, suggesting that $H_2SO_4$ could be less likely to form due to the combination of higher temperatures and lower available $SO_2$ (Hallar et al., 2016; Yu et al., 2015). $SO_2$ is not available for the entirety of the dataset, hindering the direct connection between $H_2SO_4$ precursors to the occurrence of NPF at SPL.

The CS and environmental conditions are two additional factors that can potentially explain the presence of higher aerosol concentrations during the summer and fall, despite the lack of a CCN enhancement due to NPF. The CS is a parameter that indicates how fast aerosols will condense onto pre-existing particles while also indicating how many pre-existing particles are present (Kulmala et al., 2001; Pirjola et al., 1999). Table 2 shows that CS values are highest in the summer, followed by the fall at SPL, indicating there is more pre-existing aerosol in the summer and fall than in the spring and winter. Data from the Whistler Aerosol and Cloud study, which also takes place in a montane setting in western North America, also finds that particles are more likely to grow to CCN relevant sizes when the CS is lower since there are fewer particles to react with condensable gases, a trend that is also observed in this work (Pierce et al., 2012). Because the CS is calculated before NPF initiation, these trends further suggest that aerosol transport to the site is not affecting the background particle concentrations during events. More work to analyze the relationship between CS and particle transport is required since the role the CS has on NPF is highly dependent on the conditions of a given site. Environmental conditions in the Intermountain U.S., such as wildfires, are another factor that could help explain the higher CCN concentrations present in the summer and the fall during both events and non-events since aged smoke has been observed to enhance CCN concentrations at sizes above 80 nm in the western US (Twohy et al., 2021). With wildfires becoming more frequent in the western US, CCN from wildfire emissions is expected to be a contributor to total CCN during the summer and fall months at SPL (Hallar et al., 2017). More work is needed to better understand the role that the CS and wildfires play on CCN at SPL during the summer and the fall.

The lack of a significant CCN enhancement by NPF at SPL during the summer suggests that one potential phenomenon influencing NPF, and eventually CCN concentrations, is that lower temperatures are lowering the energy barrier required for $H_2SO_4$ formation in the winter and spring (Yu et al., 2015). This suggests that an anthropogenic source of $SO_2$, similar to the powerplants upwind of SPL, is one important aspect for the occurrence of NPF events that can enhance CCN observed in the

spring and winter at SPL (Hallar et al., 2016). Other mountaintop studies that report NPF events enhancing CCN are near an anthropogenic emission source. For example, the Mt. Chacaltaya Observatory, where previous studies report 61% of events grow to CCN sizes, is located 15 km away from the city of La Paz, Bolivia (Rose et al., 2017). Mt. Tai, a mountaintop observatory in Shandong, China on the transport path of the Asian continental outflow, reports a decreased frequency of NPF events that grow to CCN sizes because of decreases in $SO_2$ concentrations over time, demonstrating the importance that $H_2SO_4$ precursors have on growing aerosols from NPF to CCN sizes (Zhu et al., 2021; Fu et al., 2008). The results from this work can be compared to other results from studies that report an enhancement of CCN due to NPF (Table 3)

## 5. Verifying the Automatic Methodology to Classify New Particle Formation

To verify the automatic classifications of NPF events, we visually classify NPF events and then compare the agreement. Figure 6 contains the total number of days classified into the four classification schemes: event (includes type 1a and type 1b events), class II event, non-event, and undefined. The agreement rate of the four classification schemes between visual and automatic classification is 51%. The automatic method classifies more days as undefined (automatic: 32.2% of days) compared to the visual classification method (visual: 14.6% of days), leading to this poor agreement rate. However, this agreement rate increases to 79% when considering the binary classification of events (type 1a event, type 1b event, class II event) and non-events (undefined, non-event).

A large source of the days classified as undefined by the automatic method are days in which 5 Gaussian maxima are not able to be calculated. These are days that are classified as events, undefined, and non-events during visual classification. Future work to improve the visual classification method should consider why specific days may not have Gaussian maxima fitted and thus could be incorrectly classified as undefined days. The automatic method has an 85% agreement with visual classification at identifying events when these undefined days due to a lack of Gaussians are removed demonstrating a generally good overall agreement. Because the automatic method analyzes the number concentrations with different metrics while visual classification looks at patterns in a colored size distribution, the automatic method may be more sensitive to small perturbations in the data. More studies utilizing automatic methodology and comparing automatic classification to visual classification will help to determine aspects where automatic classification can be improved.

Because the particle growth rate and $J_8$ values are based on calculations implemented in the automatic methodology, analysis of these variables can allow for further verification of the automatic method (Table 2). Average growth rates are the highest in the summer (10.61 nm h$^{-1}$) and lowest in the winter (4.98 nm h$^{-1}$). The spring and fall seasons display similar growth rates (spring: 6.51 nm h$^{-1}$, fall: 5.83 nm h$^{-1}$), which could indicate that the growth rate displays a seasonal cycle at SPL. Compared to Hallar et al., (2011) which utilizes visual classification methods, growth rates determined by the automatic classification are similar, albeit lower by 13.2% in the spring, higher by 16.5% in the summer, and lower by 14.1% in the winter (Hallar et

al., 2011). These results affirm that the automatic method is calculating growth rates well since there is an expected difference due to different dates considered in each study. $J_8$ values are also calculated at SPL for all seasons (Table 2). Average seasonal $J_8$ values range from 1.76 #/cm$^{-3}$ s$^{-1}$ to 11.07 #/cm$^{-3}$ s$^{-1}$, which are higher than the average seasonal values observed at SPL in 2011 ranging from 0.37 #/cm$^{-3}$ s$^{-1}$ to 1.19 #/cm$^{-3}$ s$^{-1}$ (Hallar et al., 2011). Because this study uses the

methodology of Kulmala et al., 2012 and Haller at al., 2011 uses methodology from Kulmala et al., 2004, differences between the two studies are expected since loss terms are not considered in the simplified equation used in Hallar et al., 2011 (Kulmala et al., 2004; Hallar et al., 2011; Kulmala et al., 2012).

While the $J_8$ values are higher than calculated in Hallar et al., (2011), the seasonal variation in the $J_8$ values, with the highest

observed values in the summer, indicates that our method aligns with previous work. Observations of summer NPF at SPL indicate that short bursts of particles are common in the summer which would lead to higher $J_8$ values (Yu and Hallar, 2014). The higher growth rates observed in the summer accompanied by slightly shorter event durations further supports that NPF in the summer is likely due to significant bursts rather than prolonged growth (Table 2). Since 83% of events classified by the method in the summer are class II events, the automatic method is successfully identifying these burst days and

calculates variables that are consistent with these observations.

While a comparison with the automatic methods that use deep learning-based convolution neural networks (CNN) (Joutsensaari et al., 2018; Su et al., 2022) would provide an important comparison, training the CNN would require the removal of the data used in training from consideration. For example, Su et al., 2022 requires 358 annotated days to train and only classifies class 1 (banana shaped) events while our method can also identify class II days. Joutsensaari et al., 2018

presents another option of automatic classification using deep learning but recommends 150 days per class to properly train the method for each site. The big advantage of our method compared to other automatic methods is that aspects of the statistical method can be altered to fit individual sites without having to train the method. Assuming there are enough data available, future studies focusing on using automatic methodology should attempt to use both the statistical method detailed here, and CNN based automatic methods.

**6. Conclusions**

This work at SPL marks the first time long-term, direct observations of aerosols and CCN are analyzed in North America to quantify the impact NPF events have on CCN concentrations. Findings show that NPF events significantly enhance CCN concentrations in the spring by a factor of 1.54 and in the winter by a factor of 1.36 while there is no significant

enhancement observed in the summer or fall. Type 1a and type 1b NPF events, characterized by persistent growth, are more common in the spring and winter while class II burst events are more common in the summer and fall. Lower temperatures which decrease the barrier for nucleation in the spring and winter alongside higher levels of $SO_2$ (an important $H_2SO_4$

precursor) in these seasons are likely factors that contribute to the occurrence of NPF events that eventually enhance CCN concentrations (Hallar et al., 2016; Yu et al., 2015).


An innovative aspect of this research is the implementation of two new automatic methods: one to classify NPF and another to determine the times when CCN concentrations are impacted by NPF. The automatic method to identify NPF produces an overall event frequency of 50% which compares well to event frequencies calculated by previous studies using visual classification. A comparison of the automatic classification method to visual classification produces close to an 80%

agreement showing the promise of automatic methodology to be applied in future studies. A threshold method to determine $CCN_{start}$ and a growth-based method to determine $CCN_{end}$ ensure that CCN concentrations are considered during times that particles from a given NPF event could activate as CCN. These methods are easily applicable to larger datasets making it possible to increase efficiency when comparing the effect of NPF on CCN at multiple sites.

At SPL, the presence of an anthropogenic $SO_2$ plume from upwind coal-fired powerplants during the spring and winter appears to be an important factor allowing for particles from NPF to eventually activate as CCN. Similar enhancements of CCN in remote, continental regions, such as SPL, may require an anthropogenic source of NPF precursors to grow to sizes relevant to CCN formation. These results are in contrast to previous modeling studies that find NPF reduces CCN providing a new perspective on the significant extent NPF enhances CCN concentrations in remote regions with close proximity to an

anthropogenic source of $H_2SO_4$ precursors (Sullivan et al., 2018). More studies connecting NPF to CCN in different regions across the globe will add important information and increase understanding of the climate-relevant relationship between NPF and CCN.

**Code and Data Availability**


All aerosol and CCN data will be available on the EBAS dataset. Please feel free to reach out to the corresponding author for access to relevant data and/or code.

**Author Contributions**


The manuscript was written by Hirshorn with contributions from all authors. Experiment design, data analysis, and methodology development were performed by Hirshorn, Zuromski, Rapp, Carrillo-Cardenas, and Hallar. McCubbin, Yu, and Hallar further contributed materials and analysis tools necessary for the study.

**Acknowledgments**

The authors would like to acknowledge the National Science Foundation (Grant #1951632) and the University of Utah Global Change and Sustainability Center for their support. Thank you to Maria Garcia and Dan Gilchrist for their hard work maintaining instruments at Storm Peak Laboratory.


**Competing Interests**

The authors declare no competing interests.

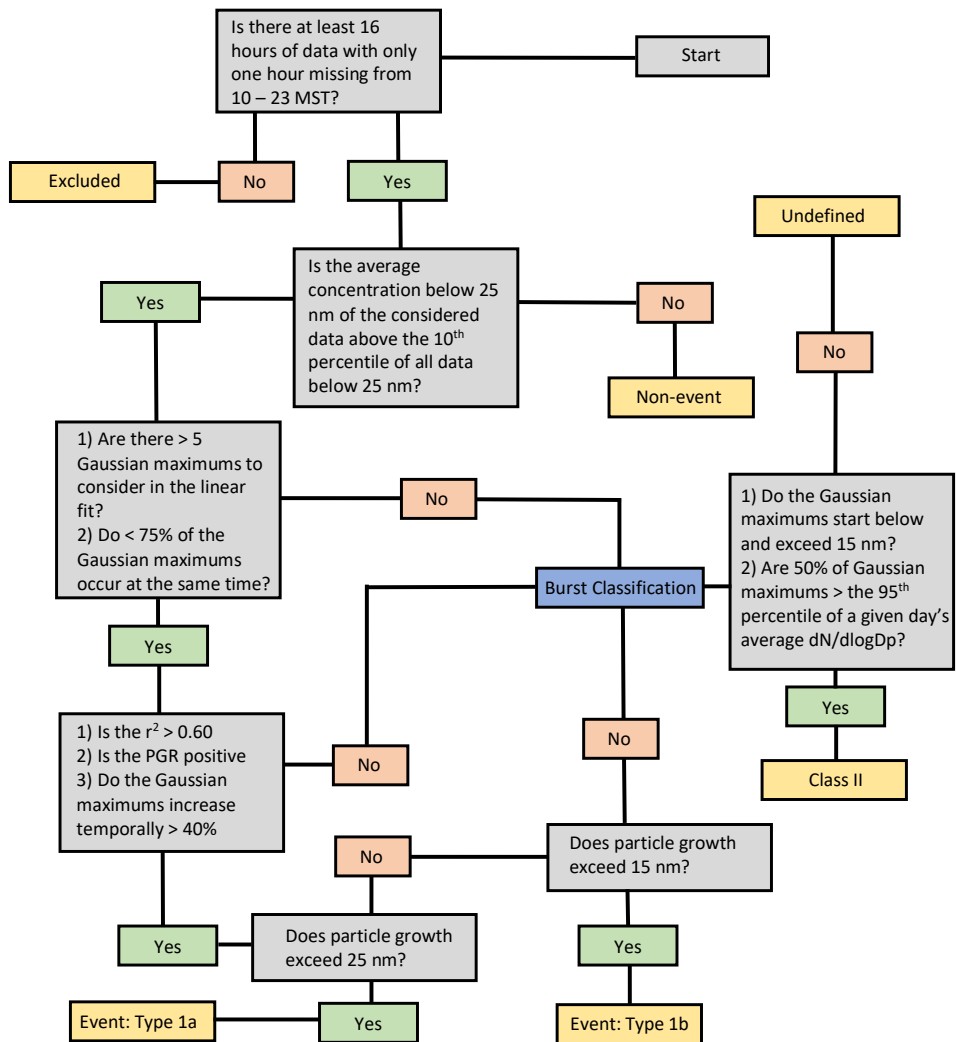


**Figure 1:** Flowchart illustrating the step-by-step process of the automatic NPF classification method.

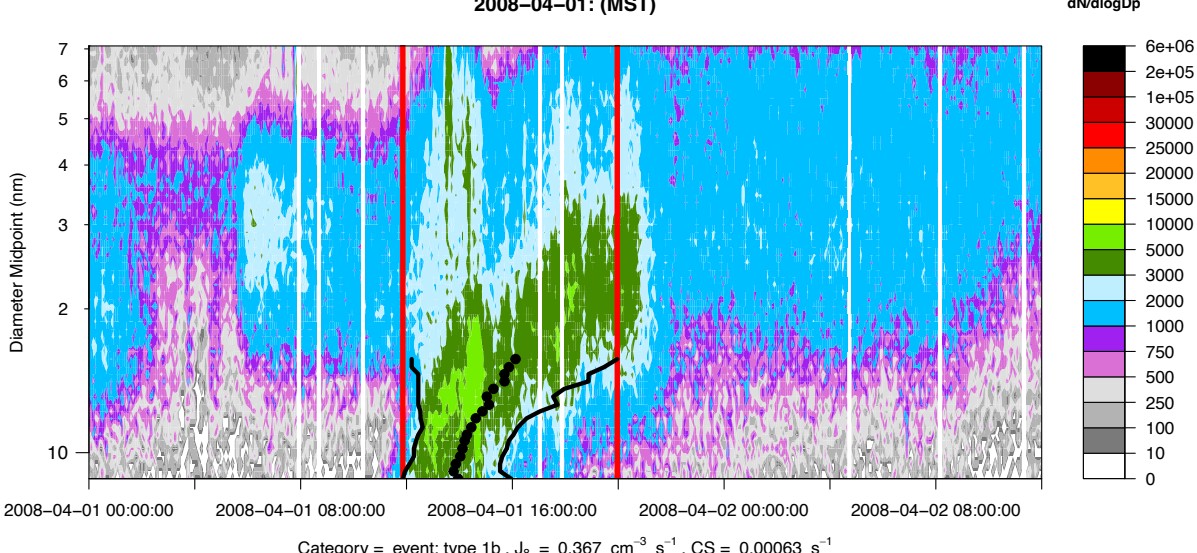

**Figure 2:** An example of a day classified as a class Ib event. Setting 15 nm as the diameter that the growth Gaussian maxima
must reach allows for this day to be classified as an event demonstrating why the threshold is set at 15 nm. Gaussian
maximums (black points) are outlined by the first-order derivative of the fitted distribution at each size (black line). The
vertical red lines denote the initiation and end times of a given event as assigned by the automated methodology

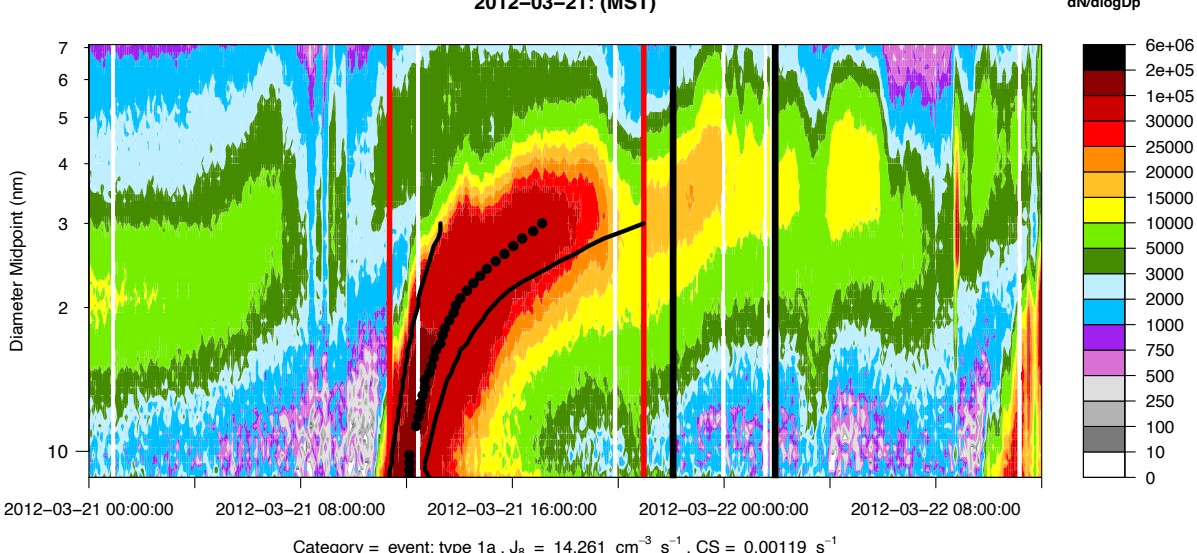

Category = event: type 1a , $J_8$ = 14.261 cm$^{-3}$ s$^{-1}$ , CS = 0.00119 s$^{-1}$

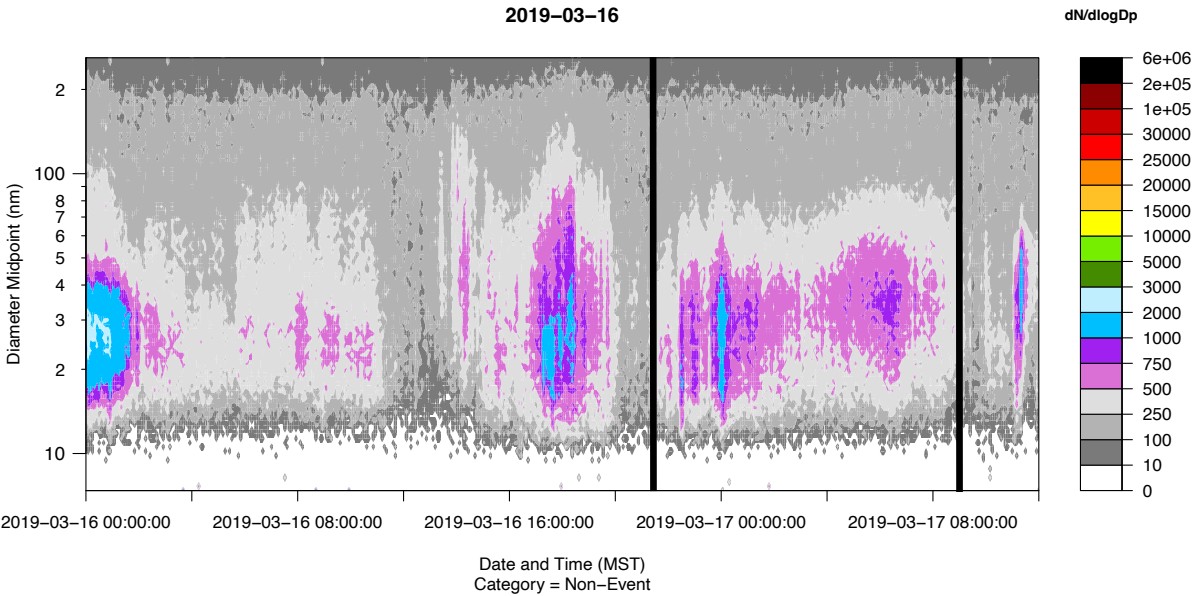

Date and Time (MST)
Category = Non-Event

**Figure 3:** Strong NPF event (top) with midpoint size bin maximums (black points), outlined by the first-order derivative of the fitted distribution at each size (black sloped lines). The vertical red lines denote the initiation and end times of a given event as assigned by the automatic methodology. A non-event (bottom) is added for comparison. The vertical black lines represent the time period when CCN is considered (CCN$_{start}$ through CCN$_{end}$) which is determined for each individual event day while the seasonal average of this period is used for comparing CCN during non-event days.

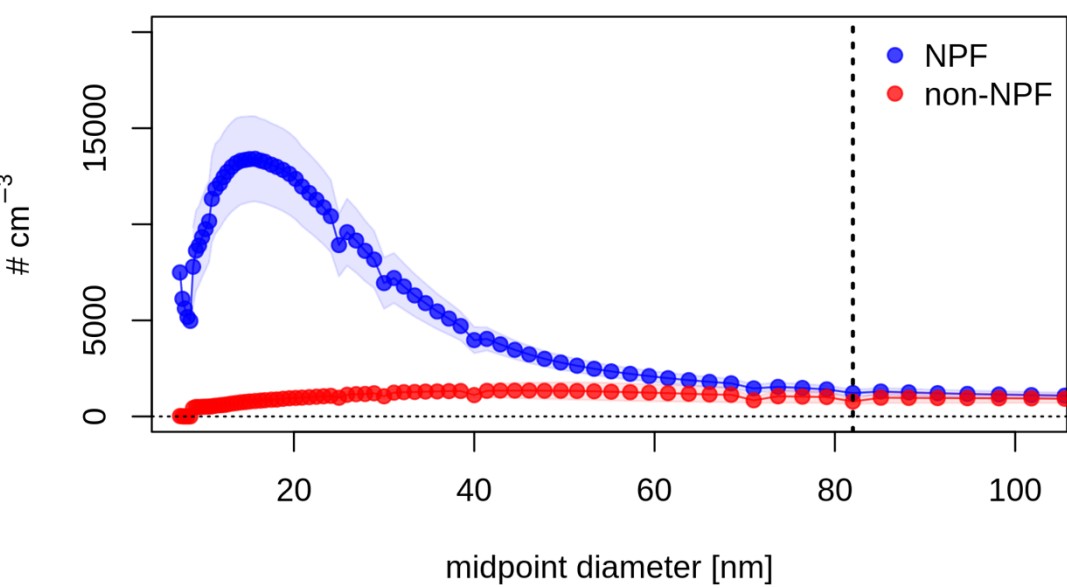

**Figure 4:** Average number of particles produced at each particle diameter for NPF events (blue) and non-events (red). NPF events produce significantly more particles at aerosol diameters below the vertical line (82.0 nm) as determined by a two-sample t-test ($p < 0.05$ indicates significance).



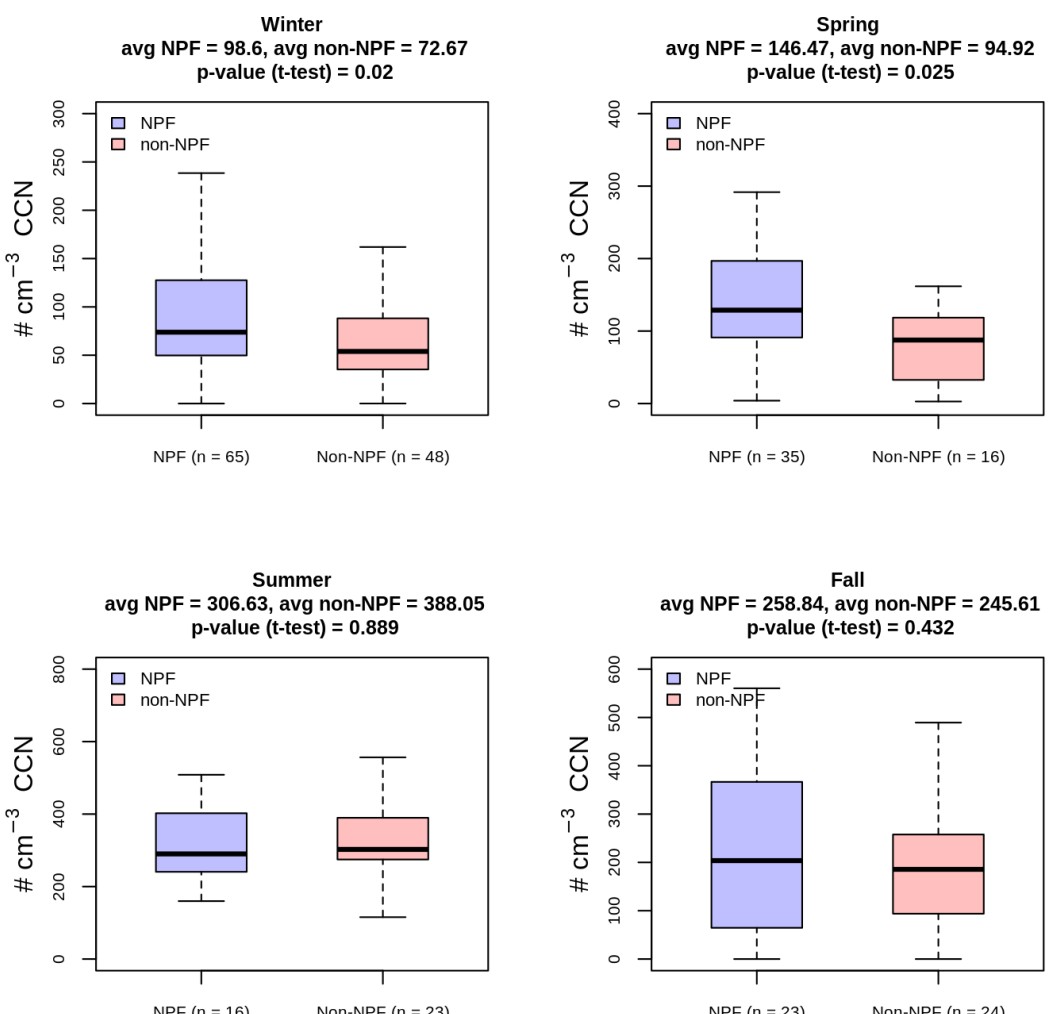

**Figure 5:** Seasonal Comparisons of average CCN number concentrations (# cm$^{-3}$) during NPF events (blue) and non-events (red). CCN concentrations during events are considered starting at the CCN$_{start}$ time and ending at the CCN$_{end}$ time of a given day. CCN is considered during non-events starting at the seasonal average of CCN$_{start}$ and ending at the seasonal average of CCN$_{end}$. Displayed p-values represent the results of a two-sample t-test with a one-sided hypothesis that NPF days would exhibit greater CCN concentrations than non-events. We interpret values below p = 0.05 to be statistically significant. P-values show that the spring and winter display statistically significant enhancements of CCN due to NPF, a trend that is absent during the summer and fall.


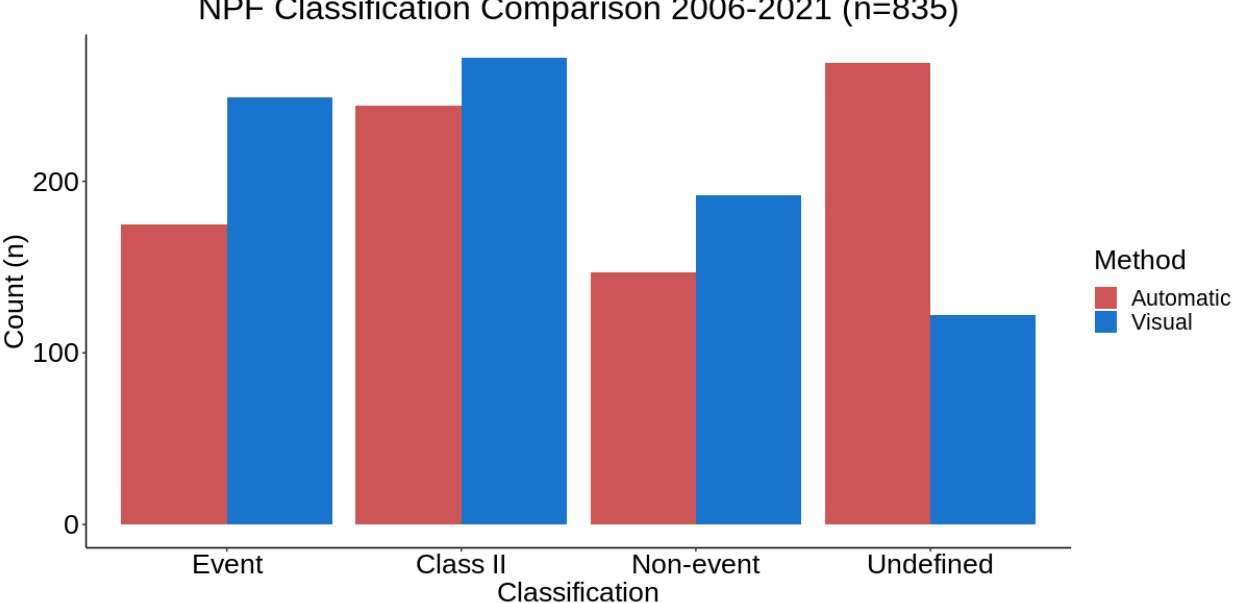

**Figure 6:** Comparisons detailing the number of days considered as a given classification category between the automatic classification method (red) and visual classification (blue). The event category includes type 1a and type 1b events.





| Variables | Spring | Summer | Fall | Winter |
|---|---|---|---|---|
| Total Days Considered | 170 | 178 | 179 | 308 |
| Type 1a Events | 29 | 10 | 20 | 36 |
| Type 1b Events | 17 | 7 | 14 | 42 |
| Class II Events | 44 | 82 | 71 | 47 |
| Undefined Events | 56 | 45 | 41 | 127 |
| Non-events | 24 | 34 | 33 | 56 |
| Total Event Frequency | 53% | 56% | 59% | 41% |
| Frequency of Type 1a/1b Events | 51% | 17% | 32% | 62% |

**Table 1:** A summary of variables related to the frequency of NPF split by season. The total event frequency considers the percentage of days that a type 1a event, type 1b event, or class II event occurs compared to an undefined or non-event day. The "Frequency of Type 1a/1b Events" row considers the percentage of all events in a given season that are persistent growth events (type 1a event or type 1b event).





| Variables | Spring | Summer | Fall | Winter |
|---|---|---|---|---|
| Average Particle Growth Rate (nm h$^{-1}$) | 6.51 ± 3.66 | 10.61 ± 5.60 | 5.83 ± 4.17 | 4.98 ± 3.06 |
| Average Formation Rate ($J_8$) (cm$^{-3}$ s$^{-1}$) | 3.51 ± 4.35 | 11.07 ± 22.35 | 1.86 ± 3.14 | 1.76 ± 2.41 |
| Average Event CS (10$^{-3}$ s$^{-1}$) | 0.90 ± 0.54 | 2.65 ± 1.51 | 1.39 ± 0.96 | 0.64 ± 0.35 |
| Average Non-event CS (10$^{-3}$ s$^{-1}$) | 0.71 ± 0.50 | 3.03 ± 1.88 | 1.40 ± 1.41 | 0.46 ± 0.43 |
| Average NPF Initiation Time (MST) | 12:40 ± 2.20 h | 13:35 ± 2.01 h | 12:44 ± 3.28 h | 12:51 ± 1.30 h |
| Average Event Duration (Hours) | 8.19 ± 5.30 | 7.52 ± 4.92 | 8.39 ± 6.11 | 8.84 ± 4.94 |
| Average CCN$_{start}$ Time (MST) | 21:02 ± 5.35 h | 16:36 ± 2.82 h | 16:49 ± 3.49 h | 20:01 ± 4.71 h |
| Average Duration of CCN Consideration (CCN$_{end}$ − CCN$_{start}$) (Hours) | 11.62 ± 5.56 | 14.13 ± 4.89 | 15.22 ± 5.22 | 12.77 ± 6.18 |

**Table 2:** Seasonal summary of variables calculated for type 1a and 1b events. Values are presented as the mean of the variable ± one standard deviation.




| Site | Authors | Environment | Time Period | NPF Frequency | Contribution of NPF to CCN |
|---|---|---|---|---|---|
| Storm Peak Laboratory, Steamboat Springs, CO, USA | Hirshorn et al., 2022 | Mountaintop | 2006 - 2021 | 50% | 1.36 enhancement in winter, 1.54 enhancement in spring |
| Mt. Chacaltaya Observatory, Bolivia | Rose et al., 2017 | Mountaintop | 2012 | Boundary layer: 48% Free troposphere: 39% | Boundary layer: 67% of events enhance CCN Free troposphere: 53% of events enhance CCN |
| Vienna, Austria | Dameto de España et al., 2017 | Urban | 2014 - 2015 | 13% | 14 days display 1.43 enhancement |
| University of Crete at Finokalia, Crete, Greece | Kalkavouras et al., 2019 | Coastal | 2008 - 2015 | 162 episodes | 1.29 – 1.77 enhancement |
| Polarstern Research Vessel near Svalbard, Norway | Kecorius et al., 2019 | Polar | 2017 | 4 events analyzed | Enhancement factor 2-5 |
| Iberian Peninsula, Spain | Rejano et al., 2021 | One urban site, one mountaintop site | 2018-2019 | Urban: N/A Mountaintop: N/A | Urban: N/A Mountaintop: 1.75 |
| 35 sites worldwide | Ren et al., 2021 | Multiple Sites Urban and Remote | Varied | N/A | Urban: 3.6 enhancement Remote: 1.8 enhancement |


**Table 3:** Details of multiple studies that find the enhancement of CCN by NPF using observational data. For a study to be included on this list, an enhancement percentage or factor of CCN due to NPF must be calculated.

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
