# Peer review of "Seasonal Significance of New Particle Formation Impacts on Cloud Condensation Nuclei at a Mountaintop Location"

_Atmospheric Chemistry and Physics, 2022_

## Author Comment (AC1)

**Hirshorn et al., Responses to Reviewer 1**

The authors of this paper would like to thank the reviewer for their insightful and constructive comments on the paper. We have carefully considered the feedback and this resulted in major improvements to the paper. Please note that references to line numbers in the author responses correspond to the new line numbers in the updated manuscript.

The authors have color coded the responses to the reviewer as follows:

Blue: A response to the reviewer.
Black: Text that is in the originally submitted manuscript.
Red: Changes that were made to the manuscript and are reflected in the updated manuscript.

**Major Comments**

**1. I suggest the authors to improve methodology section (specific comments below). Specially, it is not clear how the authors account the contribution of NPF to CCN, and only the timing of the events is presented.**

Response: Lines 279 – 289 now include information on how we consider the contribution of NPF to CCN so that methods that detail the timing of CCN consideration, as well as how we compare CCN from events to non-events, are now included. We aim to highlight how this robust approach can be used in similar studies with long-term datasets.

Addition: To compare the impact NPF events have on CCN, CCN number concentrations directly measured are considered during the time period spanning from $CCN_{start}$ to $CCN_{end}$ during valid events and non-events. An average CCN number concentration for supersaturation levels between 0.2% and 0.4% is calculated for each individual time period. These values are then averaged each season separately between events and non-events.  The goal is  to determine whether CCN concentrations are enhanced by NPF events. During long-term studies, especially at clean, remote locations like SPL, directly comparing events and non-events will result in the relative enhancement of CCN due to events at a given location. By removing the subjectivity of selecting idealized cases, we provide a more robust methodology to evaluate long-term datasets. The methodology within this paper carefully considers similar timeframes within the diel pattern with and without NPF, to look at the relative change induced by NPF. By further comparing events to non-events through a seasonal lens, we ensure that days with similar meteorological conditions are compared. By further comparing events to non-events through a seasonal lens, we ensure that days with similar meteorological conditions are compared.

**2. This manuscript presents a new methodology to classify NPF events, however it has been only applied at SPL site and validation, success ratio and/or comparison with other methods in detail are not provided. Despite it is a visual classification and can lead to human biases, Dal Maso et al 2005 has been used for years as an standardized method to classify NPF events. This methodology is presented as new, however it is based on Dal**

**Maso et al methodology. Why not comparing results in deep? This is not the first automatic method in the literature (e.g. Su et al., 2022) and no comparison, benefits or improvements are shown. Finally, the authors don't provide the procedure to calculate the GR, the formation rate is calculated with a formula that is simplified (and not correct), the diffusion coefficient is assumed to be 0.077 cm2 s-1 (this factor depends on the temperature and pressure, how representative is for SPL?) and the factor beta is also considered to be unity (why?). Kulmala et al 2012 provided guidelines to compare different NPF studies.**

Response: One aspect that was often revisited when conducting this work was the subjectivity of both visual and automatic classification. We completely agree that Dal Maso et al., 2005 is a quintessential paper in our field, but there is a reliance on expert visual classifiers that are often not available or subjective. The goal of our statistical based method here is to provide a way that long-term datasets can be classified in a more efficient manner. It would be impossible to not consider Dal Maso et al., 2005 when creating the method which is why we provide a general comparison of the results of the automatic method to visual classification in section 5 of the paper. The goal here is to highlight the comparability of two methods while also acknowledging that discrepancies will occur due to the slightly different approaches of both methods. While Dal Maso et al., 2005 has been used for a longer period of time, we do believe that the promise of a more statistical-based and efficient way to classify NPF is why it is important to develop a statistical based method.

While a comparison with the automatic methods that use convolution neural networks (Joutsensaari et al., 2018; Su et al., 2022) was considered, we did not want to train the neural network with data classified by the statistical method in our study. For example, Su et al., 2022 requires 358 annotated days to train and only classifies class 1 (banana shaped) events while our method can also identify class II days. Joutsensaari et al., 2018 presents another option of automatic classification using deep learning but recommends 150 days per class to properly train the method for each site. The big advantage of our method compared to other automatic methods is that aspects of the statistical method can be altered to fit individual sites without having to train the method.

The growth rate of an NPF event is found by finding the slope of the linear regression that is fit to the maximum Gaussians. This procedure follows the conventional growth rate equation by using the growth pattern observed in the Gaussian maximums. The equation and a more detailed explanation can now be found on lines 180-190 and in the figure below.

Addition: For days that are defined as an event, the growth rate and event start and stop times are calculated. The growth rate is determined by the following equation which uses the linear regression fit to the maximum Gaussians:

$$GR = \frac{d}{dt}\left(D_p\right) = \frac{\Delta D_p}{\Delta dt} \tag{2}$$

Because the slope of the linear regression fit of the maximum Gaussians represents particle growth over time during NPF events, this value is used when determining the growth rate . Derivatives of the linear regressions are used to determine the start and end time of events, where the start time of the event is defined by the

time of the first maximum of the first-order derivative, and the end time of the event is defined by the time of the last first-order derivative minimum. Figure 3 illustrates an example of an NPF event, and a day classified as a non-event.

[Figure]

For the $J_8$ equation, the authors acknowledge that the equation we use is simplified. To maintain consistency across studies, we use the simplified form of the equation to mirror the work of Hallar et al., 2011 which uses Kulmala et al., 2004 to justify using the equation focused on sources while not considering sinks because of the clean conditions at SPL. A reference to Hallar et al., 2011 is added on line 227 to acknowledge that we also use this paper for justifying the equation. The values we get are different than Hallar et al., 2011, but we do believe that this has to do more with the time period considered in the calculations which is now determined through

the automatic classification method. Lines 221 – 228 in the paper describe the protocol which we clearly outline to ensure the reader understands how we calculate the formation rate.

Addition: Where  $\Delta N_{8,Dmax}$ is the change in the number concentration of particles across the size distribution from about 8 nm to the maximum diameter (about 340 nm) during $\Delta t$ which is the time difference from the defined start of an event to the defined end of an event. When calculating the initial and final number concentrations, we utilize the average number concentration observed between 4 hours and 1 hour prior to NPF initiation as the initial number concentration. The final number concentration is the average number concentration from all 5-min scans taken during an event. Doing so allows for the comparison of the initial conditions of an NPF event, and aerosol formation across the entirety of a given event. We use the above formation rate equation because conditions at SPL are conducive to clean, homogenous air masses allowing for the use of the simplified version of the equation (Kulmala et al., 2004; Hallar et al., 2011).

Thank you for pointing out the error in the CS calculation. We have spent time revisiting the calculation for the diffusion coefficient and $\beta$. Below is a step by step process that details how we came up with a representative diffusion coefficient (0.13 cm$^2$s$^{-1}$) as well as how we calculate $\beta$ for each dataset:

The condensation sink in this work is calculated using the following equation (Kulmala et al., 2012):

$$CS = 2\pi D \int_0^\infty d_p \beta_m(d_p) n(d_p) dd_p = 4\pi D \sum_i \beta_i r_i N_i$$

In the equation, r is the radius of a given size bin (cm), N is the number concentration (#/cm$^3$), D is the diffusion coefficient of H$_2$SO$_4$ (cm$^2$/s), and $\beta$ is the transition regime correction. Both D and $\beta$ are calculated out for SPL.

$\beta$ can be calculated using the following equation (Kulmala et al., 2012; Tuovinen et al., 2020):

$$\beta = \frac{1 + kn}{1 + 0.377Kn + \frac{4}{3}\alpha^{-1}Kn + \frac{4}{3}\alpha^{-1}Kn^2}$$

Where $\alpha$ is the mass accommodation coefficient which is assumed to be unity in order to mirror previous work by Hallar et al., 2011. Kn is the Knudsen number which can be calculated by $\lambda_v/r$ and will be a unique value at each radius. Radius must be in units of meters in this equation for the units to work. The following equation is used to calculate $\lambda_v$:

$$\lambda_v = 3\sqrt{\frac{\pi m_v}{8kT}} * D$$

D is the diffusion coefficient of vapor (m²/s), k is the Boltzmann number (m²kg/s²K), T is the temperature in Kelvin (seasonal average temperature used for each given dataset), and $m_v$ is the molecular mass of $H_2SO_4$ (Kg).

The value of D is assumed to be 0.13 cm²s⁻¹ after using representative data for SPL (680 mb pressure, -25 to 25 degrees C temperature range) and plugging these values into the following equation (Tuovinen et al., 2021, Welty et al., 2020):

$$D = \frac{10^{-3} T^{1.75} \sqrt{\dfrac{1}{M_{air}} + \dfrac{1}{M_{vapor}}}}{p \left( \sum_{v,air}^{1/3} + \sum_{v,vapor}^{1/3} \right)^2}$$

$M_{air}$ and $M_{vapor}$ are molecular weights in (g mol⁻¹) of each respective gas. T is the absolute temperature in Kelvin, $\Sigma_{v,air}$ and $\Sigma_{v,vapor}$ are diffusivity volumes of the air, and the condensing vapor molecule. P is pressure in standard atmosphere pressure (atm).

Lines 235-241 have been updated to address how we calculate both variables and the correct values of CS are updated in the paper.

Addition:  In the equation, $r_i$ is the radius of a given size bin (cm), and $N_i$ is the number concentration (#/cm³) of the given size bin. D is the diffusion coefficient of vapor, which is assumed to be 0.13 cm²s⁻¹ for $H_2SO_4$ at SPL based on calculations using representative pressure and temperature at the site (Hanson and Eisele, 2000; Welty et al., 2020; Tuovinen et al., 2021). $\beta_m$ is calculated following the protocols of Kulmala et al., 2012 and Tuovinen et al., 2020 (Kulmala et al., 2001; Nishita et al., 2008; Hallar et al., 2011; Kulmala et al., 2012; Tuovinen et al., 2020).

Because much of the work to calculate these variables was conducted by Gerardo Carrillo-Cardenas, he has been added as a co-author to the paper.

**3. When talking about the impact of NPF to CCN concentrations, this method is not well explained and further explanations are needed. In addition, this method does not show clear advantages with those previously presented in the literature and I suggest the authors**

**to look in deep some of the issues discussed in previous works (e.g. Dameto de España et al., 2017; Rejano et al., 2021; Rose et al., 2017).**

The authors would like to first acknowledge that the work conducted in the three previous papers helped to influence the method we have created to identify when to consider CCN as being enhanced by NPF. Below we have included a short summary of each of the aforementioned studies:

- Dameto de España et al., 2017: Two years of data with 38 events that have concurrent CCN measurements. 15 days were analyzed for CCN. The study uses a CCNC and tracks the time it takes from NPF initiation to CCN relevant sizes (time period 1) and considers CCN as enhanced by NPF for a time period the same length as time period 1. In addition, an event must have consistent traffic emissions concentrations, a consistent wind direction, and a stable mixing layer height.
- Rose et al., 2017: Data throughout 2012 with 94 analyzed NPF events. The study did not have a CCNC available. To consider CCN enhancements, the study utilizes a methodology to identify times in which NPF contributes to CCN concentrations, starting from when aerosol number concentrations at 50 nm, 80 nm, and 100 nm begin to increase and ending when the maximum number concentration is observed at the respective size bin.
- Rejano et al., 2021: Two years of data at two different sites. 15 of the clearest NPF days considered that display banana growth. The study did have a CCNC; however, the $N_{CCN}$ was also estimated using aerosol properties (optical and from the size distribution).

When comparing our study to Dameto de España et al., 2017, Rejano et al., 2021, and Rose et al., 2017, a notable difference between the data available in each study is that our dataset spans 15 years compared to either 1 or 2 years in the aforementioned studies. Because of the length of our dataset, we aimed to define a series of thresholds that could easily be applied to size distributions to identify times starting when enough particles from an NPF event hit CCN relevant sizes ($CCN_{start}$) and ending when the growth of an event tapers off ($CCN_{end}$). The advantages of this method specifically compared to the above studies is that these thresholds can be applied to large datasets to identify the time which to consider CCN efficiently. Furthermore, only the size distribution is considered when identifying the CCN consideration times in our work which allow for a higher number of days to be considered.

Of the above methods, Dameto de España et al., 2017's paper appears to be the most similar because we both base the start time of CCN consideration on a time when particles reach CCN sizes. Because their work took place in Vienna, Austria there are multiple protocols to remove days influenced by urban emissions which could taint the NPF event. Due to SPL's remote location, we do not consider protocols in the same way as Dameto de España et al... Rather, we analyze the long-term growth of events and end CCN consideration when a given NPF event stops growing.

We acknowledge that there are differences between NPF events within our study, but we do believe that the high number of days considered by the study will ensure that any tainted days,

which are less likely to occur in a remote location, will have a minimal effect on the data allowing for a comprehensive analysis of the enhancement of CCN due to NPF events.

Based on the above analysis and discussion, we believe our study has the following advantages making it important to get into the body of literature:

- Start and end times of CCN consideration are both considered based on the growth of particles in a given particle size distribution.
- By considering a higher number of days and splitting data into seasonal categories, we create a comprehensive and honest analysis of NPF's enhancement of CCN. We consider NPF events (type 1) some of which are clear type 1a events and others are weaker type 1b events. However, both occur at SPL and by considering both we can narrow in on the true enhancement of CCN.
- Determining CCN consideration times can be quickly done for large datasets eliminating the need for visual analysis of each day.

**1) We can assume that all the particles >100nm will act as CCN, however not all particles below 100nm come from NPF events, so you can explore some subtracting method to account for that?**

**3) Free troposphere conditions will probably reduce the number of NPF events, and boundary layer conditions will lead to higher event frequency, why not using same atmospheric conditions to subtract the effect from lower sizes? 4) SMPS measures from 8 to 340 nm, if above 100nm we have the largest contribution to CCN concentrations, which errors have the increase factors that you present here?**

These are both great points that the reviewer brings up. Given the clean conditions of SPL, the consideration of event vs. non-event days will help to address this concern. While particles above 100 nm will be present during both events and non-events, the comparison of the two will highlight the differences between the two classification categories while minimizing the effect of days that could negatively impact the comparison. Particles above 100 nm are considered within the aerosol size distribution for each individual day when determining when to consider CCN concentrations. We ensure that this detail is present on lines 250-251. The high number of days considered will result in a normalized comparison of events (type 1a and type 1b) vs. non-events. The methodology within this paper carefully considers similar timeframes within the diel pattern with and without NPF, to look at the relative change induced by NPF. Additional lines describing the importance of the comparison of events and non-event is added on 283 – 289.

Lines 253-254: For days classified as type 1a events and type 1b events, the start time of CCN consideration ($CCN_{start}$) is the first time after the start of an NPF event that 25% of all particles in a given scan (ranging from 8 nm to about 340 nm) are above 40 nm.

Addition: During long-term studies, especially at clean, remote locations like SPL, directly comparing events and non-events will result in the relative enhancement of CCN due to events at a given location. By removing the subjectivity of selecting idealized cases, we provide a more

robust methodology to evaluate long-term datasets. The methodology within this paper carefully considers similar timeframes within the diel pattern with and without NPF, to look at the relative change induced by NPF. By further comparing events to non-events through a seasonal lens, we ensure that days with similar meteorological conditions are compared. By further comparing events to non-events through a seasonal lens, we ensure that days with similar meteorological conditions are compared.

**2) SPL is a mountain site, the difference between event and non-event days will probably be affected by the transport from lower altitudes, I suggest to add some results/discussion about free troposphere conditions, influence from boundary layer, and the differences during event and non-event days.**

Diel patterns of aerosols at Storm Peak Laboratory have previously been attributed to almost daily transitions of boundary layer and free tropospheric air masses at Storm Peak Laboratory (Borys and Wetzel, 1997). Since radiosondes are not commonly launched within the near vicinity of Storm Peak Laboratory, we have no representative potential temperature profiles and we therefore generally refer to nighttime air masses as regional air masses, although they may represent free tropospheric air as shown in previous studies (Borys and Wetzel, 1997, Borys et al., 1986).  Generally data suggest a minimum concentrations of condensation nuclei (CN) in the early mornings which are considered background tropospheric concentrations (Lowenthal et al., 2002, Richardson et al., 2007; Obrist et al., 2008).  Thus, the methodology within this paper carefully considers similar timeframes within the diel pattern with and without NPF, to look at the relative change induced by NPF.

References:

R.D. Borys, M.A. Wetzel; Storm Peak Laboratory: a research, teaching and service facility for the atmospheric sciences; Bulletin of the American Meteorological Society, 78 (1997), pp. 2115-2123.

R.D. Borys, D.H. Lowenthal, K.A. Rahn; Contributions of Smelters and Other Sources to pollution of sulfate at a mountaintop site in northwestern Colorado. Acid Deposition in Colorado – Local Versus Long-Distance Transport into the State; CIRA, Colorado State University, Ft. Collins, CO (1986); pp. 167–174.

D.H. Lowenthal, R.D. Borys, M.A. Wetzel; Aerosol distributions and cloud interactions at a mountaintop laboratory; Journal of Geophysical Research, 107 (2002), p. 4345, 10.1029/2001JD002046

M.S. Richardson, P.J. DeMott, S.M. Kreidenweis, D.J. Cziczo, E.J. Dunlea, J.L.Jimenez, D.S. Thomson, L.L. Ashbaugh, R.D. Borys, D.L. Westphal, G.S. Casuccio, T.L. Lersch; Measurements of heterogeneous ice nuclei in the western United States in springtime and their relation to aerosol characteristics; Journal of Geophysical Research – Atmospheres, 112 (D2) (2007), p. D02209

Obrist, D., Hallar, A. G., McCubbin, I., Stephens, B. B., & Rahn, T. (2008). Atmospheric mercury concentrations at Storm Peak Laboratory in the Rocky Mountains: Evidence for long-range transport from Asia, boundary layer contributions, and plant mercury uptake. *Atmospheric Environment*, *42*(33), 7579-7589.

**4. The abstract doesn't provide new findings. 1) NPF occurs 50% of all days (if you use a new method to classify NPF events and you compare results with previous methods, it could be a highlight); 2) Events with persistent growth are common in spring and winter; 3) NPF enhances CCN by a factor 1.36, that combined with previous work at SPL, suggests the enhancement of CCN?. These three new findings pointed in the abstract could be results of a measurement report (not for a research paper). The results 1) and 2) have been already reported previously by Hallar et al. 2011.**

Response: We are glad that the reviewer brought this point up because by rewriting the abstract, we feel that the main points of the study are more clearly highlighted ensuring that the reader will understand the main findings of this work by just reading the abstract. Additions are reflected on lines 16 – 24.

Addition:  Using the new automatic method to classify NPF, we find that NPF occurs on 50% of all days considered in the study from 2006 to 2021 demonstrating consistency with previous work at SPL. NPF significantly enhances CCN during the winter by a factor of 1.36 and the spring by a factor of 1.54, which, when combined with previous work at SPL, suggests the enhancement of CCN by NPF occurs on a regional scale. We confirm that events with persistent growth are common in the spring and winter, while burst events are more common in the summer and fall.  A visual validation of the automatic method was performed in the study. For the first time, results clearly demonstrate the significant impact of NPF on CCN in montane North American regions and the potential for widespread impact of NPF on CCN.

**Minor Comments**

**L24-70 – There is a lack of references that have previously investigated the impact of NPF on CCN concentrations, some of them on mountain sites and combining PNSD and CCN and/or using monodisperse (e.g., Kalkavouras et al., 2019; Dameto de España et al., 2017; Kalkavouras et al., 2019; Kalivitis et al. 2015; Kecorius et al. 2019; Rejano et al., 2021; Rose et al. 2017) and some NPF studies in mountain sites.**

Response: All the references suggested above are now included in an expanded introduction paragraph (Lines 74-93). The papers listed above that were not included in the first version of the manuscript were re-reviewed to determine additional places throughout the paper where they could be included. We appreciate that the above literature suggestions make the reference list more complete.

Addition: In an effort to increase the number of observational studies relating NPF to CCN, previous studies, both with and without a CCNC, have developed various methodologies to determine the time period in which observed CCN concentrations can be attributed to the occurrence of an NPF event (Kalivitis et al., 2015; Kalkavouras et al., 2017; Dameto de España et al., 2017; Rose et al., 2017; Kalkavouras et al., 2019; Kecorius et al., 2019; Rejano et al., 2021; Ren et al., 2021) Similar to the methodology of Rose et al., (2017), detailed above, Kalkavouras et al., (2017) estimates CCN by finding particle concentrations above the minimum size required for aerosols to activate as CCN and then considers the environmental supersaturation when estimating how many aerosols in a given distribution could act as CCN. This approach further calculates the droplet number and considers how supersaturation, chemical composition, and updraft velocity may impact the cloud droplet number (Kalkavouras et al., 2017). An evolution of this approach by Kalkavouras et al., (2019) calculates the relative dispersion of CCN at different supersaturations and considers CCN times when the relative dispersion is higher than initial conditions before a CCN event (Kalkavouras et al., 2019). This method was further employed at 35 different sites around the globe, both urban and remote, to determine the impact NPF has on CCN concentrations (Ren et al., 2021). Kecorius et al., 2019 utilized a CCNC in the arctic to analyze CCN enhancements by fitting a slope to CCN measurements starting when aerosol formation rates increased and ending when an air mass shift occurred (Kecorius et al., 2019). In another study utilizing a CCNC in Vienna, Austria, Dameto de España et al., (2017) considers CCN number concentrations for a time period that occurs after, and for the same duration as the time difference between NPF initiation and when particles reach CCN relevant sizes (Dameto de España et al., 2017). When it comes to determining the time period that NPF may impact CCN for long term datasets, the methodology should not only efficiently and independently (without using CCN observations) ensure that aerosols are growing to CCN sizes but also needs to consider the growth patterns of individual NPF events to accurately determine when NPF stops contributing to CCN.

**L91-99 – CPC model? Do you routinely calibrate the instrumentation? Please, include both information.**

Response: Information regarding the specific CPC model and routine instrument upkeep are now included on lines 115-118.

Addition: To measure aerosols at SPL, we use a TSI Inc. (Shoreview, MN) Scanning Mobility Particle Sizer (SMPS) 3936 (with a TSI 3010 Condensation Particle Counter [CPC]) for particles with diameters between 8 and 340 nm that scans every 5 minutes. Data is collected on a log normal scale with particle diameter on a log scale and time on a normal scale. The instrument is periodically shipped back to TSI Inc. for routine maintenance and calibrations.

**L104, L107, 108 – These references are mainly based on the methodology presented by Dal Maso et al. (2005).**

Response: Additional references to Dal Maso et al., 2005 are now included on lines 135 and 137. Although building off the work of Dal Maso et al., 2005; these references are included because of the progress they make to expand on the number of classification categories which reflects the categories used in this paper.

Addition: In an effort to improve the visual classification process proposed by Dal Maso et al., 2005, studies split events into subcategories to provide more specific classifications detailing whether particle growth is sustained during a given day, or if the given day exhibits a burst of particles (Hirsikko et al., 2007; Kulmala et al., 2012; Boy et al., 2008; Svenningsson et al., 2008; Dal Maso et al., 2005).

**Figure 1 – "Is the average concentration below 25 nm above the 10th percentile of all data?" What means? All data serie, 10th percentile of total particle concentration of that 5min data, daily concentrations?**

Response: For all days considered by the classification, the average concentration of particles below 25nm is calculated. If a given day's average falls below the $10^{th}$ percentile of all data, it is assumed that there is not an NPF event due to the lack of particles below 25 nm in the size distribution.

To clarify this step in the process, the box in the flowchart now reads "Is the average concentration below 25 nm of the considered data above the $10^{th}$ percentile of all data below 25 nm?" This detail is also clarified on Lines 157-159 of the edited manuscript.

Addition: For days  where the average particle concentration below 25 nm is above the $10^{th}$ percentile of all data considered, the maximum of the Gaussians is calculated at each size bin.

**L127-136 – The Gaussians are calculated following the equation 1, however, I can not see the diameter parameter. Are you using lognormal distribution?  The time index, where is that index? "k" is the maximum aerosol number concentration" for each of the modes I guess?. Please check some references as Huusein et al. 2008 (equations) or Hussein et al. 2005 (DO-FIT algorithm) and rewrite this explanation, difficult to understand which fit method are you applying. In addition, 5 different maximum points? 5 different Gaussians? why that number?**

Response: We have made adjustments to the equation and the explanation on lines 159-170 in an effort to make the equation easier to interpret. The new equation reads as follows:

$$f(t \mid k, \mu, \sigma) = k e^{-\frac{(t-\mu)^2}{2\sigma^2}}, \quad k = max\left(\frac{dN}{dlogD_p}\right)$$

The new format of the equation clarifies that "k" indicates the max number concentration at a given $D_p$ and replaces x with "t" which represents the time at which the normalized maximum number concentration at $D_p$ occurs.

We have further added an explanation to clarify that the non-linear least squares estimate we use is the same one detailed in Bates and Watts 1988. Another aspect that is clarified is the role of $D_p$ (a given diameter midpoint in the size bin). The equation itself does not rely on a $D_p$ value, but rather considers the number concentration over time at a given $D_p$. It is this distribution that the maximum Gaussian is determined for. We also want to clarify that aerosol data follows a log-normal distribution; with the progression of $D_p$ being on a log scale. Line 117 earlier in the paper now clarifies the scale of the data.

5 different Gaussian maximum points was set as the threshold because the authors deemed anything lower than 5 points an inadequate number of points to run a linear regression. This is important because the linear regression helps calculate growth rate and determine whether the growth of a given day is strong enough to be considered an event. Line 168 now includes a clarification that we consider 5 different Gaussian maximum points.

Additions: The normal distributions were fit by solving for the non-linear least-squares estimates using the R programming language (Equation 1)  which considers the particle size distribution at each diameter to return the time that corresponds to the maximum concentration at that given diameter (Bates and Watts, 1988). In the equation, "k" is the maximum aerosol number concentration,  "t" is the time index where the normalized maximum at $D_p$ occurs, "µ" is the mean aerosol concentration, and "σ" is the corresponding standard deviation:

$$f(t \mid k, \mu, \sigma) = k e^{-\frac{(t-\mu)^2}{2\sigma^2}}, \quad k = max\left(\frac{dN}{dlogD_p}\right)$$

$$(1)$$

The derived time index represents the time at which the maximum of the peak fitted particle size distributions occurs for each  value of $D_p$. For data where at least 5 different Gaussian maximum points are calculated, a linear regression is fit to these maxima allowing for further analysis of growth over the course of an event (Lehtinen and Kulmala, 2003).

**Figure 2 – specify what the black lines indicate (and red ones).**

Response: The caption for Figure 2 is rewritten to address what the black and red lines represent.

Addition: An example of a day classified as a class Ib event. Setting 15 nm as the diameter that the growth Gaussian maxima must reach allows for this day to be classified as an event demonstrating why the threshold is set at 15 nm. Gaussian maximums (black points) are outlined by the first-order derivative of the fitted distribution at each size (black line). The vertical red lines denote the initiation and end times of a given event as assigned by the automated methodology

**Figure 3 – the authors identify the bottom figure as a weak event, why? There is no new mode appearing below 25 nmor growing.**

Response: Thank you for pointing out the error. This was a typo on our part and the bottom plot was always intended to represent a non-event day. The caption for Figure 3 now reflects this change.

Addition: Strong NPF event (top) with midpoint size bin maximums (black points), outlined by the first-order derivative of the fitted distribution at each size (black sloped lines). The vertical red lines denote the initiation and end times of a given event as assigned by the automatic methodology. A  non-event (bottom) is added for comparison. The vertical black lines represent the time period when CCN is considered (CCN$_{start}$ through CCN$_{end}$) which is determined for each individual event day while the seasonal average of this period is used for comparing CCN during non-event days.

**Figure 4 – please use log-scale (or log-log)**

Response: The authors appreciate the suggestion and we created plots of this relationship on a log scale. After seeing these plots, the authors do believe that the normal scale demonstrates the relationship the best out of all the plots. If we were considering diameters above 100 nm in the specific plot, we would put the x-axis in a log scale. However, because the diameter range we are evaluating is below 100 nm, we believe the normal scale plot is a valid way to demonstrate the data.

---

## Author Comment (AC2)

**Hirshorn et al., Responses to Reviewer 2**

The authors of this paper would like to thank the reviewer for their insightful and constructive comments on the paper. We have carefully considered the feedback. Based on this feedback, we have made significant changes that have improved the work. Please note that references to line numbers below in the author responses correspond to the new line numbers in the updated manuscript.

The authors have color coded the responses to the reviewer as follows:

Blue: A response to the reviewer.
Black: Text that is in the originally submitted manuscript.
Red: Changes that were made to the manuscript and are reflected in the updated manuscript.

1. **Line 86 – BVOCs can impact aerosol formation and growth.**

Response: The fact that BVOCs impact aerosol growth as well as aerosol formation is now reflected on line 111 of the updated manuscript.

Addition: SPL is located above a mixed forest allowing for the emission of a variety of different biogenic volatile organic compounds (BVOCs) that can impact aerosol formation and growth (Amin et al., 2012).

2. **Line 93-94 – Define Level 1, Level 2 and EBAS.**

Response: A description highlighting the difference between level 1 and level 2 EBAS data is now included on lines 120 – 125. EBAS is the database of EMEP (European Monitoring and Evaluation Programme) EMEP is the co-operative program for monitoring and evaluation of the long-range transmission of air pollutants in Europe.  EBAS is operated by the Norwegian Institute for Air Research.  We credit the Norwegian Institute for Air Research on line 125.

Addition:  SMPS data from SPL are now available on the EBAS database (database of European Monitoring and Evaluation Programme) including level 1 data, which maintains 5 min time resolution while removing invalid values and calibrations, as well as level 2 data which presents hourly averages and quantifies atmospheric variability. Level 1 SMPS data is used in this study. The goal of EBAS data is to store long-term atmospheric science datasets and provide standards for quality assurance, thus rigorous standards for data quality are implemented to any data admitted to EBAS (Norwegian Institute for Air Research).

3. **Line 127 – what is "ample"?**

Response: In this case, ample means that the average number concentration below 25 nm for a given day falls above the $10^{th}$ percentile of all considered data. To better clarify what the word ample means, the sentence on lines 157 – 159 was edited.

Addition: For days  where the average particle concentration below 25 nm is above the 10th percentile of all data considered, the maximum of the Gaussians is calculated at each size bin.

**4. Line 181 – ΔN8,Dmax rather then N**

Response: We have corrected the value to $\Delta N_{8,Dmax}$ and the sentence on lines 221 - 223 now reflects this change.

Addition: Where  $\Delta N_{8,Dmax}$ is the change in the number concentration of particles across the size distribution from about 8 nm to the maximum diameter (about 340 nm) during $\Delta t$ which is the time difference from the defined start of an event to the defined end of an event.

**5. Line 212 – It is unclear here why you bother with CCN for non-event days. Presumably, it is because you use this as a reference point for comparison of CCN impacted by events. The reason should be made clear here.**

Response: The authors agree that the reasons for why events are compared to non-events needs to be highlighted. The goal with this comparison is to truly highlight the potential CCN enhancement of NPF events compared to days with no traces of NPF. By conducting this comparison on such a large dataset, we believe that this reference is suitable as it is specific to location, season, and diurnal cycles, . Multiple additions to the manuscript have been made below to better highlight the reason for comparisons.

Addition (Lines 257 – 261): We consider CCN concentrations during non-events to determine if NPF events result in an enhancement of CCN.  Sunlight is generally necessary for NPF and growth; therefore, it is important to consider the variations in the seasonal diurnal cycle and obtain  one unique value of $CCN_{start}$ for each season that accurately represents the time that NPF impacts the site  during each season (Hallar et al., 2011).

Addition (Lines 279 – 289): To compare the impact NPF events have on CCN, CCN number concentrations directly measured are considered during the time period spanning from $CCN_{start}$ to $CCN_{end}$ during valid events and non-events. An average CCN number concentration for supersaturation levels between 0.2% and 0.4% is calculated for each individual time period. These values are then averaged each season separately between events and non-events. The goal is to determine whether CCN concentrations are enhanced by NPF events. During long-term studies, especially at clean, remote locations like SPL, directly comparing events and non-events will result in the relative enhancement of CCN due to events at a given location. By removing the subjectivity of selecting idealized cases, we provide a more robust methodology to evaluate long-term datasets. The methodology within this paper carefully considers similar timeframes within the diel pattern with and without NPF, to look at the relative change induced by NPF. By further comparing events to non-events through a seasonal lens, we ensure that days with similar meteorological conditions are compared. By further comparing events to non-events through a seasonal lens, we ensure that days with similar meteorological conditions are compared.

**6. Line 213 – NPF and growth are impacted by sunlight as well as many other factors. Rather than "NPF and growth are largely impacted by sunlight, therefore…", I suggest something like "Sunlight is generally necessary for NPF and growth, and therefore…"**

Response: The suggestion has been implemented on lines 256-258 of the edited manuscript.

Addition: Sunlight is generally necessary for NPF and growth; therefore, it is important to consider the variations in the seasonal diurnal cycle and obtain  one unique value of $CCN_{start}$ for each season that accurately represents the time that NPF impacts the site  during each season (Hallar et al., 2011).

**7. Line 214-215 – Here you state that you "obtain four different values of CCNstart". I assume, but may be wrong, that you mean one for each season, but not four for each event, which is how it sounds. Clarify please.**

Response: We have clarified that one value of $CCN_{start}/CCN_{end}$ is determined for each season. This change is reflected on lines 256-258 as well as an additional clarification on lines 274-275.

Addition on 259-261: Sunlight is generally necessary for NPF and growth; therefore, it is important to consider the variations in the seasonal diurnal cycle and obtain  one unique value of $CCN_{start}$ for each season that accurately represents the time that NPF impacts the site  during each season (Hallar et al., 2011).

Addition on 277-278: Four different values of $CCN_{end}$, one for each season, are determined when finding $CCN_{end}$ values for non-events.

**8. Line 223-224 – It is written to sound like this is novel, yet surely it is obvious that you must allow a long enough time.**

Response: We confirm that considering NPF overnight at our remote site is not a novel aspect of our work. However, the goal of this sentence is to show the reader that this aspect of NPF is not just considered by our method, but also why we decide to consider general particle growth patterns after the most intense period of NPF. This will hopefully clarify to the reader, and potential users of this method, that we have considered this aspect of NPF in the determination of our CCN consideration times.

**9. Lines 248- 250 - Is 53% significantly lower than 56% in this case?**

Response: 53% is not significantly lower than 56% in this case. To emphasize this point, the sentence on lines 305-306 was edited.

Addition: Spring (53%) and winter (41%) display similar but slightly lower event frequencies than the summer and fall at SPL.

**10.   Lines 260-263 - What is the method of production of particles on non-event days, and are they truly new particles or just the result of a meteorological change, such as a developing boundary layer (since you are at a mountain site)?**

Response: This is a great point and something that needs to be addressed to confirm that NPF days at SPL truly do represent a contribution of new particles compared to non-event days. In Hallar et al., 2016, a nano-SMPS was used to show that NPF events are accompanied with a burst of particles that is observed as low as 5 nm. Below is an example of a size distribution from the nano-SMPS:

[Figure]

By observing these particles at such low sizes, we have added confidence that observed particles during event days are newly formed and thus NPF, not solely transported through a meteorological change. Data from the nano-SMPS is only available for 1 year in our data collection period which is why this analysis is not included in this long term study. However, to emphasize that there has been previous work to highlighting that these particles are indeed due to NPF, a sentence on lines 324-325 was added:

Addition: Previous work at SPL has shown that during NPF events, particles as low as 5 nm are observed alongside events demonstrating that particles observed during NPF originate from nucleation (Hallar et al., 2016).

**11.   Line 265 – I suggest changing "produce" to "indicate".**

Response: This change has been made on Line 322.

Addition: Above 82 nm, days with NPF events do not  indicate more particles than non-events, which suggests any enhancements in CCN due to NPF events are likely due to particles below 82 nm.

**12.   Lines 266-267 - I'm having trouble understanding the apparent use of non-events as a reference.  Why is it necessary to use non-events this way.  Are you assuming that days with events would be the same as non-event days, IF there was no event?  Presumably, one of the triggers for the event could be that the aerosol immediately preceding the event was particularly low in number and size (i.e., low CS).  Would the non-event reference be appropriate in that case?  Also, it is possible that the aerosol sampled on non-event days may well have been influenced by NPF at some time in its history.  Would you clarify your reasons behind this, please?**

Response: The justification for putting the section that details Figure 4 in the paper is to highlight that more particles below 82 nm in diameter are produced during events which emphasizes a quantifiable difference between events and non-events at SPL that could potentially impact CCN.

Given that there is a 15-year dataset and multiple event and non-event days for each season, we believe that comparing type 1a and type 1b NPF events to non-events is the best way to compare the contribution of CCN from days with an NPF event compared to days where this phenomenon is completely absent. At a mountaintop observatory, there is no such thing as a perfect NPF event or a perfect non-event. However, the purpose of having such a long dataset is that days with an anomaly will be less prominent in the comparison as they are averaged out by the more characteristic NPF events and non-events. This ensures that the highest number of days possible are included in this comparison providing the best overview of the relationship between NPF events and non-events at SPL.

One aspect of the code that helps ensure a quality comparison is the undefined classification category. The undefined classification category includes days where there is evidence of elevated aerosol concentrations but do not exhibit the growth of an event. This category helps to capture days without NPF events that have higher aerosol concentrations ensuring that the non-event category represents clean conditions at SPL absent of any burst like behavior that could skew the comparison.

Furthermore, as shown in Hallar et al., (2011) and here, we do not see a significant difference between the CS before NPF events and on days without NPF events throughout the entire dataset.  Thus, in this clean remote region, it appears the CS does not solely drive the occurrence of an NPF event.

**13.    Line 283 - Here, the summer result again suggests the "non-event" may not be a suitable reference.**

Response: This response builds on the response to the above comment. The summer result showing more CCN during non-event days compared to event days was not what we were expecting to find. However, this comparison is an important result that highlights why we conduct the comparison from a seasonal lens. As detailed in the paper, the summertime displays different conditions (wildfires, higher temperatures leading to BL fluctuations, organics) than other seasons. What this shows us is that while days with NPF events produce CCN, they are not a significant contributor due to other factors that can influence days without an NPF event. The seasonal comparison helps us conduct an accurate comparison while identifying the seasons where this relationship may not be true as well as the factors that influence this relationship.

**14.    Line 303 – What about condensable organics?  Do higher temperatures somehow inhibit growth by organics?  How would your CCN number concentrations react if organics played a major role in growth of the newly formed particles, compared with sulphate?  What do you think are the main precursors leading to particles growth at SPL?**

Response: We found it best to format the response by responding specifically to one question at a time.

**What about condensable organics? Do higher temperatures somehow inhibit growth by organics?**

The potential role of organics was left out of the paper because the relationship between temperature and particle growth by organics is complex and non-linear. This relationship is highly dependent on the organic species that is observed and was explored by Stolzenburg et al., 2018 (DOI: https://doi.org/10.1073/pnas.1807604115) finding that higher temperatures can lead to increased reaction rates and concentrations of highly oxidized organic molecules. However, lower temperatures can help decrease volatility resulting in less oxidized species becoming able to condense. Thus, there is not only a reliance on temperature, but also a reliance on the degree a molecule is oxidized. While obtaining measurements of organics that can detail the degree of oxidation would be highly beneficial, it is outside of the scope of this work since we do not have that data available.

**How would your CCN number concentrations react if organics played a major role in growth of the newly-formed particles, compared with sulphate?**

The authors believe that this is a question that is beyond the scope of this work. To answer this question, we would need access to the specific types of organics that are observed at SPL during each season, $SO_2$ measurements to compare the general organic:sulfate ratio, and aerosol composition measurements. This is a fantastic question that could be a great idea for future work but we do not currently have the measurements to conduct a clear comparison.

**What do you think are the main precursors leading to particles growth at SPL?**

We believe that the main precursor leading to particle growth at SPL is the presence of $H_2SO_4$ that is formed due to $SO_2$ emissions from the powerplants upwind. Lower temperatures that allow for nucleation to occur are another important factor. UV radiation and westerly winds (location of powerplants) have also been proven to lead to more favorable conditions of NPF. The role of organics has yet to be investigated in depth. The impact organics may have on NPF at SPL is highly dependent on the type of organics observed and will require further work to make a conclusion.

**15. Lines 310-333 - You have more bursts in summer and fall that suggests H2SO4 is being produced. Are you suggesting that it ends up condensing on existing larger particles, thereby improving their ability to act as CCN? In the first paragraph here, you say that a reduced CS is important for your SPL observations of NPF, and that is shown in Table 2. However, then you lead off the second paragraph stating that one phenomenon is influencing NPF and CCN on event days at SPL in the summer: temperature. I feel that this is misleading without a detailed analysis of the many things that might affect NPF: including, temperature, CS, SO2 concentrations, irradiance, available condensable species. In relatively clean environments with low concentrations of precursors (e.g., Arctic), a low CS can be to be a trigger for NPF. Higher concentrations of SO2 superimposed on regions with a low CS will result in higher number concentrations of NPF. Higher SO2 concentrations in regions with a higher CS will tend to have lower NPF. In other words, the CS is important, along with other factors. Unless you can present more evidence or a stronger argument that temperature is the primary factor controlling NPF here, I think the focus on one factor may be misleading and the discussion should be made a little more objective.**

Response: The authors agree that it is important to ensure that the discussion reflects the work done in the paper and is not overly objective or accidentally misleading. To address this concern, language in the discussion has been added and/or changed to be less objective and reflect the scope of the work. Lines 345-397

[revised manuscript text omitted]

**16.   The paper would be helped by putting the present results in perspective through a simple tabled comparison with other estimates of the contribution from NPF to CCN in the literature.**

Response: This is a great idea. Table 3 in the edited manuscript now includes results from other studies that detail the enhancement of CCN due to NPF events. We limited the table to observational studies that explicitly report an enhancement factor of CCN due to NPF. A reference to the table is made on lines 396-397.

Addition: The results from this work can be compared to other results from studies that report an enhancement of CCN due to NPF (Table 3)

Additions:

| Site | Authors | Environment | Time Period | NPF Frequency | Contribution of NPF to CCN |
|---|---|---|---|---|---|
| Storm Peak Laboratory, Steamboat Springs, CO, USA | Hirshorn et al., 2022 | Mountaintop | 2006 - 2021 | 50% | 1.36 enhancement in winter, 1.54 enhancement in spring |
| Mt. Chacaltaya Observatory, Bolivia | Rose et al., 2017 | Mountaintop | 2012 | Boundary layer: 48% Free troposphere: 39% | Boundary layer: 67% of events enhance CCN Free troposphere: 53% of events enhance CCN |
| Vienna, Austria | Dameto de España et al., 2017 | Urban | 2014 - 2015 | 13% | 14 days display 1.43 enhancement |
| University of Crete at Finokalia, Crete, Greece | Kalkavouras et al., 2019 | Coastal | 2008 - 2015 | 162 episodes | 1.29 – 1.77 enhancement |
| Polarstern Research Vessel near Svalbard, Norway | Kecorius et al., 2019 | Polar | 2017 | 4 events analyzed | Enhancement factor 2-5 |
| Iberian Peninsula, Spain | Rejano et al., 2021 | One urban site, one mountaintop site | 2018-2019 | Urban: N/A Mountaintop: N/A | Urban: N/A Mountaintop: 1.75 |

| 35 sites worldwide | Ren et al., 2021 | Multiple Sites Urban and Remote | Varied | N/A | Urban: 3.6 enhancement Remote: 1.8 enhancement |
|---|---|---|---|---|---|

**Table 3:** Details of multiple studies that find the enhancement of CCN by NPF using observational data. For a study to be included on this list, a enhancement percentage or factor of CCN due to NPF must be calculated.

**17.  Lines 397-398 – Scale is an important factor when comparing remote regions with areas of strong anthropogenic influence.  There has been a reasonable amount of work examining NPF in the Arctic (a remote region).  I think before concluding with this statement, you should look at some related literature: for example, Nieminen et al. (ACP, 2018; https://doi.org/10.5194/acp-18-14737-2018); Abbatt et al. (ACP, 2019; https://doi.org/10.5194/acp-19-2527-2019 ), and references therein.**

Response: The authors thank the reviewer for providing these papers. We agree that the statement made should be reworded after examining the literature to place an emphasis that SPL's remote location does display differences from remote locations in polar regions. One of the big takeaways is that NPF in polar regions relies heavily on DMS from the ocean (natural influence) while our continental site might rely on power plants (anthropogenic influence). The sentence on Lines 461-463 was changed.

Addition: Similar enhancements of CCN in remote, continental regions, such as SPL, may require an anthropogenic source of NPF precursors to grow to sizes relevant to CCN formation.

---

## Author Response (AR2)

**Hirshorn et al., Responses to Editor**

The authors of this paper would like to thank the editor for reviewing the reviewer responses and providing suggestions on how to prepare the paper for publication. We have carefully considered the feedback. Based on this feedback, we have made appropriate changes to address remaining concerns. Please note that references to line numbers below in the author responses correspond to the new line numbers in the updated manuscript.

The authors have color coded the responses to the reviewer as follows:

Blue: A response to comments provided by the editor/reviewer. Black: Text that is in the originally submitted manuscript. Red: Changes that were made to the manuscript and are reflected in the updated manuscript.

**1. About comment#2 and authors response.**

a) We can agree that statistical methods are better than visualization method, however if the visualization method has been the standard, better comparison is needed and in section 5 you need to extend your analysis/discussion. Actually, the figure 6 presents 2006-2021 comparison with just n=835 days. Have the authors used ~3years data in a period of 15 years, why then the figure have 2006-2021 title? In addition, the advantage against other automatic methods presented in the answers is not really clear for the reviewer, but at least some discussion need to be included in the manuscript.

Addressing figure 6, the authors would like to acknowledge that the data considered in this study spans 2006-2021 which is why the label is included in the title. There are periods of time when the SMPS was offline due to maintenance or an instrumental issue and thus data was not collected to ensure quality. Furthermore, some days are removed from consideration if they do not meet the data quality requirements described in section 2.2 of the paper:

Lines 139-144: The first step of the automatic classification method is to ensure the availability of SMPS level 1 data. Although NPF events can span multiple days, we consider daily data (0:00 – 23:59 MST) as well as the first 12 hours (0:00 – 12:00 MST) of the next day to ensure the consideration of an NPF event doesn't prematurely end if growth continues overnight. 5-minute SMPS data is only considered if the first 24-hour period meets the following conditions: there are at least 16 hours of data present, and the period between 10:00 - 23:00 MST (the times in which NPF is most common at SPL) has less than 1 hour of data missing.

In addition, the authors would like to provide a plot that details the availability of SMPS data over the course of the study:

Regarding the additional comparison to other methods, the authors agree that an in-depth comparison will add to the paper and needs to be a subject of future work. To get access to automatic methods (Joutsensaari et al., 2018; Su et al., 2022) based on machine learning the authors would need to train methods using data in this study. As a result, we would be unable to reuse the data. An additional comparison paragraph was added to the paper:

Lines 446-454: While a comparison with the automatic methods that use deep learning based convolution neural networks (CNN) (Joutsensaari et al., 2018; Su et al., 2022) would provide an important comparison, training the CNN would require the removal of the data used in training from consideration. For example, Su et al., 2022 requires 358 annotated days to train and only classifies class 1 (banana shaped) events while our method can also identify class II days. Joutsensaari et al., 2018 presents another option of automatic classification using deep learning but recommends 150 days per class to properly train the method for each site. The big advantage of our method compared to other automatic methods is that aspects of the statistical method can be altered to fit individual sites without having to train the method. Assuming there are enough data available, future studies focusing on using automatic methodology should attempt to use both the statistical method detailed here, and CNN based automatic methods.

b) The reviewer acknowledge the answer but can not agree. In order to maintain consistency with a simplified/wrong equation, the authors can use that formula to compare with Hallar et al. (2011) but not to provide new data. The equation used by the authors is used in Hallar et al. (2011) and Kulamala et al. (2004), however this formula does not account for losses. The authors stated that "because of the clean conditions at SPL" they keep using the Hallar et al. (2011) formula, however, the equation for formation rate (Kulmala et al., 2012) does not really depend on clean or not clear conditions of the site, depends of the losses by coagulation, condensation and instrumental losses (this term we can omit). Check Kulamala et al. (2012) for equations. For clean environments the GR factor of the formation rate could even be larger than the factor  $\Delta N/\Delta Dp$ . I will not accept a manuscript presenting a new methodology that uses an old/wrong/simplified formula. Same for  $\alpha$ , i will only accept the value of unity just in case the authors demonstrate it has a minor impact on the CS values (not just because to be consistent with Hallar et al. 2011).

**J8 Values:**

Before responding to this comment the authors would like to thank the reviewer for pointing out the necessity of including loss terms in the equation and thank the editor for allowing us the time to properly calculate these terms for our paper.

In Kulmala et al., 2012, the following equation is used to address the particle formation rate  $(J_{d_n})$ :

$$J_{d_p} = \frac{dN_{d_p}}{dt} + CoagS_{d_p} * N_{d_p} + \frac{GR}{\Delta d_p} * N_{d_p} + S_{losses}$$
(1)

 $\frac{dN_{dp}}{dt}$  denotes the time evolution of the particle number size concentration,  $N_{dp}$ . The rest of the terms highlight the relative losses for aerosol formation.  $CoagS_{dp}$  is the coagulation sink in the size range of  $d_p$ , and  $\Delta d_p + d_p$ , where  $d_p$  is defined as the aerosol diameter and  $\Delta d_p$  is the range in particle diameters. GR is the particle growth rate (in nm/s and  $S_{losses}$  are additional losses. Since  $S_{losses}$  encompasses losses, such as instrument losses, that are negligible in observational work, this term is the one term of the Kulmala et al., 2012 equation that is not included in our calculation (Casquero-Vera et al., 2020).

We use this equation to calculate a particle formation rate at an aerosol diameter of 8 nm  $(J_8)$  using an aerosol diameter size range  $(\Delta d_p)$  of 8 – 25 nm. To calculate the additional loss terms, Lehtinen et al., 2007 created a relationship relating the condensation sink (*CS*) and the  $CoagS_{d_p}$  using a power-law dependence that is used in Kulmala et al., 2012:

$$CoagS_{d_p} = CS * \left(\frac{d_p}{0.71}\right)^m \tag{2}$$

Where *m* is a constant equal to -1.6. Moreover, to appropriately address the  $CoagS_{\Delta d_p}$  term in equation (1), a diameter size range ( $\Delta d_p$ ) is introduced to equation (3):

$$CoagS_{\Delta d_p} = CS * \left(\frac{\Delta d_p}{0.71}\right)^m \tag{3}$$

We consider the whole particle size distribution (8 nm - 333.8 nm) for the CS and coagulation sink to more accurately account for potential particle losses. By using the new equation, which we have high confidence in this method, we calculate new values of J8 for the paper and address these changes in the methodology:

Additions lines 214-229: The  $J_8$  value for an event is defined by the formation rate equation (Kulmala et al., 2004; Kulmala et al., 2012):

$$J_8 = \frac{\Delta N_{8,D_{max}}}{\Delta t} + CoagS_{d_p} * N_{d_p} + \frac{GR}{\Delta d_p} * N_{d_p}$$
(3)

Where  $\Delta N_{8,Dmax}$  is the change in the number concentration of particles across the considered size distribution from about 8 nm to 25 nm the maximum diameter (about 340 nm) during  $\Delta t$  which is the time difference from the defined start of an event to the defined end of an event. When calculating the initial and final number concentrations, we utilize the average number concentration observed between 4 hours and 1 hour prior to NPF initiation as the initial number concentration. The final number concentration is the average number concentration from all 5-min scans taken during an event. Doing so allows for the comparison of the initial conditions of an NPF event, and aerosol formation across the entirety of a given event. The additional loss terms in the equation represent loss to the coagulation sink, and loss due to growth out of the size range (Kulmala et al., 2012). The entire size distribution measured by the SMPS is used when calculating the coagulation sink loss term (Casquero-Vera et al., 2020). We use the above formation rate equation because conditions at SPL are conducive to clean, homogenous air masses allowing for the use of the simplified version of the equation (Kulmala et al., 2004; Hallar et al., 2011).

Lines 431 - 436: Average seasonal J8 values range from 1.76 #/cm-3 s-1 to 11.07 #/cm-3 s-1, which are higher than the average seasonal values observed at SPL in 2011 ranging from 0.37 #/cm-3 s-1 to 1.19 #/cm-3 s-1 (Hallar et al., 2011). Because this study uses the methodology of Kulmala et al., 2012 and Haller at al., 2011 uses methodology from Kulmala et al., 2004, differences between the two studies are expected since loss terms are not considered in the simplified equation used in Hallar et al., 2011 (Kulmala et al., 2004; Hallar et al., 2011; Kulmala et al., 2012). Because the determination of start and end times differs between visual and automatic elassification methods, these lower J8 values may have to do with the longer time considered in an NPF event.

Edits to Table 2:

| Average Formation Rate (J 8 ) (cm - | $0.23 \pm 0.22$ | $0.33 \pm 0.51$ | <del>0.12 ± 0.15</del> 1.86 ± | <del>0.17 ± 0.21</del> 1.76 |
|-----------------------------------------------------------|-----------------|-----------------|-------------------------------|-----------------------------|
| $^{3}$ s -1 )                                  | $3.51 \pm 4.35$ | $11.07\pm22.35$ | 3.14                          | ± 2.41                      |

**CS: Mass accommodation coefficient:**

The mass accommodation coefficient ( $\alpha$ ) that is used in this study, is highlighted in the Fuchs–Sutugin correction coefficient ( $\beta_m$ ) within the condensation sink (*CS*) equation in Tuovinen et al., 2021.  $\beta_m$  is depicted as:

$$\beta_m = \frac{1+Kn}{1+\left(\frac{4}{3\alpha}+0.337\right)Kn+\frac{4}{3\alpha}Kn^2}$$
(4)

In Kulmala et al., 2012, the following equation for the Fuchs-Sutugin correction coefficient ( $\beta_i$ ) was used:

$$\beta_i = \frac{1+Kn}{1+1.677Kn+1.333Kn^2} \tag{5}$$

It is important to note that equation (6) used in Kulmala et al. 2012, is the same as equation (5) that is in Tuovinen et al., 2021, but with the assumption of  $\alpha = 1$ . Thus, we continue to assume that the value for mass accommodation coefficient ( $\alpha$ ) is unity, in this study.

Additional changes: Both Figures 2 and 3 are edited to show accurate J8 and CS values.

c) The authors provide on comment#2 a figure that presents a Gaussian of dN/dlogD vs Dp. However, the comment to L127-136 (page 11 of authors response) indicate that the Gaussians are time dependent? Please clarify if for the GR the authors are using the "Maximum-concentration method" or the "Log-normal distribution function method" (Kulmala et al. 2012). How you choose the start and end time/diameters of the Gaussians The authors want to clarify that the figure in comment two is a conceptual figure used to illustrate how the calculations of a single Gaussian at a given size bin occur. The Gaussian calculations find the given time at which the maximum Gaussian occurs at a single size bin. The authors detail this calculation of individual Gaussians in the following portion of the manuscript on lines 153 - 164:

The normal distributions were fit by solving for the non-linear least-squares estimates using the R programming language (Equation 1) which considers the particle size distribution at each diameter to return the time that corresponds to the maximum concentration at that given diameter (Bates and Watts, 1988). In the equation, "k" is the maximum aerosol number concentration, "t" is the time index where the normalized maximum at  $D_p$  occurs, " $\mu$ " is the mean aerosol concentration, and " $\sigma$ " is the corresponding standard deviation. This equation is used for the calculation of individual maximum Gaussians at each size bin:

$$f(t \mid k, \mu, \sigma) = k e^{-\frac{(t-\mu)^2}{2\sigma^2}}, \quad k = max\left(\frac{dN}{dlogD_p}\right)$$
(1)

The derived time index represents the time at which the maximum of the peak fitted particle size distributions occurs for each value of  $D_p$ . For data where at least 5 different Gaussian maximum points are calculated, a linear regression is fit to these maxima allowing for further analysis of growth over the course of an event (Lehtinen and Kulmala, 2003).

Once the Gaussians are calculated, the growth rate can be determined based on where the Gaussians are positioned related to time. This method is most similar to the log-normal function distribution method of calculating growth rate but fits the growth rate using the position of the maximum gaussians.

Lines 181 - 183: Because the slope of the linear regression fit of the maximum Gaussians represents particle growth over time during NPF events, this value is used when determining the growth rate. This method is most similar to the log-normal function fitting method of calculating growth rate but finds the growth rate by fitting a linear regression to the maximum Gaussians.

Page 8 on answers' document: "Thus, the methodology within this paper carefully considers similar timeframes within the diel pattern with and without NPF, to look at the relative change induced by NPF". You consider time frames for event and non-event days, but background conditions are the same? The occurrence of NPF at mountain sites is triggered by an increase condensable vapors that usually is accompanied of an increase on particle concentrations. Thus, particle concentrations (and CCN) usually is larger during NPF events, not because this particles come

Thank you for providing this clarification, the authors better understand what the editor and reviewer are asking. When crafting the answer on page 8, the authors did consider two aspects detailed within the work of Sellegri et al., 2019. First, at high altitudes over 1,000 meters, transport of condensable vapors can be accompanied by particles. However, this same study lists SPL as an exception where NPF is associated with low-surface area of pre-existing particles. To analyze this relationship, the condensation sink in this study is calculated for times before NPF initiation during events, and similar representative times during non-events.

To address that the transport of pre-existing particles could be an error, the following was added to the manuscript:

Lines 285 - 289: At other high-altitude mountaintop sites around the globe, this approach could have sources of error since NPF can be associated with the transport of both condensable vapors and pre-existing aerosol that could become CCN (Sellegri et al., 2019). However, SPL seems to be an exception to this rule since previous observations of NPF show association with lower existing particle surface areas which allows for a more direct comparison of events and non-events (Hallar et al., 2011; Sellegri et al., 2019).

Lines 380 - 383: Because the CS is calculated before NPF initiation, these trends further suggest that aerosol transport to the site is not affecting the background particle concentrations during events. More work to analyze the relationship between CS and particle transport is required since the role the CS has on NPF is highly dependent on the conditions of a given site which is why it is important to report CS values.

Figure 4. Please provide the log and log-log scales at least on the answer to the reviewer. I would like to see that figure on that scale. The Non-NPF events line is not clearly seen.

The authors have provided copies of the different plots that were generated when crafting the response to reviewer 1. We still believe the normal-normal scale is the best way to observe the difference in figure 4. The figures can be viewed below:

Normal scale (proposed plot):